# Two Time-scale Off-Policy TD Learning: Non-asymptotic Analysis over Markovian Samples

**Tengyu Xu**
Department of Electrical and Computer Engineering
The Ohio State University
xu.3260@osu.edu

**Shaofeng Zou**
Department of Electrical Engineering
University at Buffalo, The State University of New York
szou3@buffalo.edu

**Yingbin Liang**
Department of Electrical and Computer Engineering
The Ohio State University
liang.889@osu.edu

## Abstract

Gradient-based temporal difference (GTD) algorithms are widely used in off-policy learning scenarios. Among them, the two time-scale TD with gradient correction (TDC) algorithm has been shown to have superior performance. In contrast to previous studies that characterized the non-asymptotic convergence rate of TDC only under identical and independently distributed (i.i.d.) data samples, we provide the first non-asymptotic convergence analysis for two time-scale TDC under a non-i.i.d. Markovian sample path and linear function approximation. We show that the two time-scale TDC can converge as fast as $\mathcal{O}(\frac{\log t}{t^{2/3}})$ under diminishing stepsize, and can converge exponentially fast under constant stepsize, but at the cost of a non-vanishing error. We further propose a TDC algorithm with blockwisely diminishing stepsize, and show that it asymptotically converges with an arbitrarily small error at a blockwisely linear convergence rate. Our experiments demonstrate that such an algorithm converges as fast as TDC under constant stepsize, and still enjoys comparable accuracy as TDC under diminishing stepsize.

## 1 Introduction

In practice, it is very common that we wish to learn the value function of a *target* policy based on data sampled by a different *behavior* policy, in order to make maximum use of the data available. For such off-policy scenarios, it has been shown that conventional temporal difference (TD) algorithms [24, 25] and Q-learning [33] may diverge to infinity when using linear function approximation [2]. To overcome the divergence issue in off-policy TD learning, [27, 26, 17] proposed a family of gradient-based TD (GTD) algorithms, which were shown to have guaranteed convergence in off-policy settings and are more flexible than on-policy learning in practice [18, 23]. Among those GTD algorithms, the TD with gradient correction (TDC) algorithm has been verified to have superior performance [17] [9] and is widely used in practice. To elaborate, TDC uses the mean squared projected Bellman error as the objective function, and iteratively updates the function approximation parameter with the assistance of an auxiliary parameter that is also iteratively updated. These two

parameters are typically updated with stepsizes diminishing at different rates, resulting the two time-scale implementation of TDC, i.e., the function approximation parameter is updated at a slower time-scale and the auxiliary parameter is updated at a faster time-scale.

The convergence of two time-scale TDC and general two time-scale stochastic approximation (SA) have been well studied. The asymptotic convergence has been shown in [4, 6] for two time-scale SA, and in [26] for two time-scale TDC, where both studies assume that the data are sampled in an identical and independently distributed (i.i.d.) manner. Under non-i.i.d. observed samples, the asymptotic convergence of the general two time-scale SA and TDC were established in [14, 36].

All the above studies did not characterize how fast the two time-scale algorithms converge, i.e, they did not establish the non-asymptotic convergence rate, which is specially important for a two time-scale algorithm. In order for two time-scale TDC to perform well, it is important to properly choose the relative scaling rate of the stepsizes for the two time-scale iterations. In practice, this can be done by fixing one stepsize and treating the other stepsize as a tuning hyper-parameter [9], which is very costly. The non-asymptotic convergence rate by nature captures how the scaling of the two stepsizes affect the performance and hence can serve as a guidance for choosing the two time-scale stepsizes in practice. Recently, [8] established the non-asymptotic convergence rate for the projected two time-scale TDC with i.i.d. samples under diminishing stepsize.

- *One important open problem that still needs to be addressed is to characterize the* **non-asymptotic** *convergence rate for two time-scale TDC under* **non-i.i.d.** *samples and diminishing stepsizes, and explore what such a result suggests for designing the stepsizes of the fast and slow time-scales accordingly. Existing method developed in [8] that handles the non-asymptotic analysis for i.i.d. sampled TDC does not accommodate a direct extension to the non-i.i.d. setting. Thus, new technical developments are necessary to solve this problem.*

Furthermore, although *diminishing* stepsize offers accurate convergence, *constant* stepsize is often preferred in practice due to its much faster error decay (i.e., convergence) rate. For example, empirical results have shown that for *one* time-scale conventional TD, constant stepsize not only yields fast convergence, but also results in comparable convergence accuracy as diminishing stepsize [9]. However, for *two* time-scale TDC, our experiments (see Section 4.2) demonstrate that constant stepsize, although yields faster convergence, has much bigger convergence error than diminishing stepsize. This motivates to address the following two open issues.

- *It is important to theoretically understand/explain why constant stepsize yields large convergence error for two-time scale TDC. Existing non-asymptotic analysis for two time-scale TDC [8] focused only on the diminishing stepsize, and does not characterize the convergence rate of two time-scale TDC under constant stepsize.*

- *For two-time scale TDC, given the fact that constant stepsize yields large convergence error but converges fast, whereas diminishing stepsize has small convergence error but converges slowly, it is desirable to design a new update scheme for TDC that converges faster than diminishing stepsize, but has as good convergence error as diminishing stepsize.*

In this paper, we comprehensively address the above issues.

## 1.1 Our Contribution

Our main contributions are summarized as follows.

We develop a novel non-asymptotic analysis for two time-scale TDC with a single sample path and under non-i.i.d. data. We show that under the diminishing stepsizes $\alpha_t = c_\alpha/(1+t)^\sigma$ and $\beta_t = c_\beta/(1+t)^\nu$ respectively for slow and fast time-scales (where $c_\alpha, c_\beta, \nu, \sigma$ are positive constants and $0 < \nu < \sigma \le 1$), the convergence rate can be as large as $\mathcal{O}(\frac{\log t}{t^{2/3}})$, which is achieved by $\sigma = \frac{3}{2}\nu = 1$. This recovers the convergence rate (up to $\log t$ factor due to non-i.i.d. data) in [8] for i.i.d. data as a special case.

We also develop the non-asymptotic analysis for TDC under non-i.i.d. data and constant stepsize. In contrast to conventional one time-scale analysis, our result shows that the training error (at slow time-scale) and the tracking error (at fast time scale) converge at different rates (due to different condition numbers), though both converge linearly to the neighborhood of the solution. Our result also characterizes the impact of the tracking error on the training error. Our result suggests that TDC

under constant stepsize can converge faster than that under diminishing stepsize at the cost of a large training error, due to a large tracking error caused by the auxiliary parameter iteration in TDC.

We take a further step and propose a TDC algorithm under a blockwise diminishing stepsize inspired by [35] in conventional optimization, in which both stepsizes are constants over a block, and decay across blocks. We show that TDC asymptotically converges with an arbitrarily small training error at a blockwisely linear convergence rate as long as the block length and the decay of stepsizes across blocks are chosen properly. Our experiments demonstrate that TDC under a blockwise diminishing stepsize converges as fast as vanilla TDC under constant stepsize, and still enjoys comparable accuracy as TDC under diminishing stepsize.

From the technical standpoint, our proof develops new tool to handle the non-asymptotic analysis of bias due to non-i.i.d. data for two time-scale algorithms under diminishing stepsize that does not require square summability, to bound the impact of the fast-time-scale tracking error on the slow-time-scale training error, and the analysis to recursively refine the error bound in order to sharpening the convergence rate.

## 1.2 Related Work

Due to extensive studies on TD learning, we here include only the most relevant work to this paper.

**On policy TD and SA.** The convergence of TD learning with linear function approximation with i.i.d samples has been well established by using standard results in SA [5]. The non-asymptotic convergence have been established in [4, 12, 30] for the general SA algorithms with martingale difference noise, and in [7] for TD with i.i.d. samples. For the Markovian settings, the asymptotic convergence has been established in [31, 28] for TD($\lambda$), and the non-asymptotic convergence has been provided for projected TD($\lambda$) in [3] and for linear SA with Markovian noise in [13, 22, 21]. For linear SA with dynamic Markovian noise, the non-asymptotic analysis of on-policy SARSA under non-i.i.d. samples was recently studied in [37].

**Off policy one time-scale GTD.** The convergence of one time-scale GTD and GTD2 (which are off-policy TD algorithms) were derived by applying standard results in SA [27, 26, 17]. The non-asymptotic analysis for GTD and GTD2 have been conducted in [16] by converting the objective function into a convex-concave saddle problem, and was further generalized to the Markovian setting in [32]. However, such an approach cannot be generalized for analyzing two-time scale TDC that we study here because TDC does not have an explicit saddle-point representation.

**Off policy two time-scale TDC and SA.** The asymptotic convergence of two time-scale TDC under i.i.d. samples has been established in [26, 17], and the non-asymptotic analysis has been provided in [8] as a special case of two time-scale linear SA. Under Markovian setting, the convergence of various two time-scale GTD algorithms has been studied in [36]. The non-asymptotic analysis of two time-scale TDC under non-i.i.d. data has not been studied before, which is the focus of this paper.

General two time-scale SA has also been studied. The convergence of two time-scale SA with martingale difference noise was established in [4], and its non-asymptotic convergence was provided in [15, 20, 8, 6]. Some of these results can be applied to two time-scale TDC under i.i.d. samples (which can fit into a special case of SA with martingale difference noise), but not to the non-i.i.d. setting. For two time-scale linear SA with more general Markovian noise, only asymptotic convergence was established in [29, 34, 14]. In fact, our non-asymptotic analysis for two time-scale TDC can be of independent interest here to be further generalized for studying linear SA with more general Markovian noise.

Two concurrent and independent studies were posted online recently, which are related to our study. [10] provided a non-asymptotic analysis for two time-scale linear SA under the non-i.i.d setting, in which both variables are updated with constant stepsize. In contrast, our study provides the convergence rate for the case with the two variables being updated by the stepsizes that diminish at different rates, and hence our analysis technique is very different from that in [10]. Another study [11] proposed an interesting approach to analyze the convergence rate of TD learning in the Markovian setting via a Markov jump linear system. Such an approach, however, cannot be applied directly to study the two time-scale TD algorithm that we study here.

## 2 Problem Formulation

### 2.1 Off-policy Value Function Evaluation

We consider the problem of policy evaluation for a Markov decision process (MDP) $(\mathcal{S}, \mathcal{A}, \mathsf{P}, r, \gamma)$, where $\mathcal{S} \subset \mathbb{R}^d$ is a compact state space, $\mathcal{A}$ is a finite action set, $\mathsf{P} = \mathsf{P}(s'|s,a)$ is the transition kernel, $r(s,a,s')$ is the reward function bounded by $r_{\max}$, and $\gamma \in (0,1)$ is the discount factor. A stationary policy $\pi$ maps a state $s \in \mathcal{S}$ to a probability distribution $\pi(\cdot|s)$ over $\mathcal{A}$. At time-step $t$, suppose the process is in some state $s_t \in \mathcal{S}$. Then an action $a_t \in \mathcal{A}$ is taken based on the distribution $\pi(\cdot|s_t)$, the system transitions to a next state $s_{t+1} \in \mathcal{S}$ governed by the transition kernel $\mathsf{P}(\cdot|s_t, a_t)$, and a reward $r_t = r(s_t, a_t, s_{t+1})$ is received. Assuming the associated Markov chain $p(s'|s) = \sum_{a \in \mathcal{A}} p(s'|s,a)\pi(a|s)$ is ergodic, let $\mu_\pi$ be the induced stationary distribution of this MDP, i.e., $\sum_s p(s'|s)\mu_\pi(s) = \mu_\pi(s')$. The value function for policy $\pi$ is defined as: $v^\pi(s) = \mathbb{E}[\sum_{t=0}^\infty \gamma^t r(s_t, a_t, s_{t+1})|s_0 = s, \pi]$, and it is known that $v^\pi(s)$ is the unique fixed point of the Bellman operator $T^\pi$, i.e., $v^\pi(s) = T^\pi v^\pi(s) := r^\pi(s) + \gamma \mathbb{E}_{s'|s} v^\pi(s')$, where $r^\pi(s) = \mathbb{E}_{a,s'|s} r(s,a,s')$ is the expected reward of the Markov chain induced by policy $\pi$.

We consider policy evaluation problem in the off-policy setting. Namely, a sample path $\{(s_t, a_t, s_{t+1})\}_{t \geq 0}$ is generated by the Markov chain according to the behavior policy $\pi_b$, but our goal is to obtain the value function of a target policy $\pi$, which is different from $\pi_b$.

### 2.2 Two Time-Scale TDC

When $\mathcal{S}$ is large or infinite, a linear function $\hat{v}(s, \theta) = \phi(s)^\top \theta$ is often used to approximate the value function, where $\phi(s) \in \mathbb{R}^d$ is a fixed feature vector for state $s$ and $\theta \in \mathbb{R}^d$ is a parameter vector. We can also write the linear approximation in the vector form as $\hat{v}(\theta) = \Phi\theta$, where $\Phi$ is the $|\mathcal{S}| \times d$ feature matrix. To find a parameter $\theta^* \in \mathbb{R}^d$ with $\mathbb{E}_{\mu_{\pi_b}} \hat{v}(s, \theta^*) = \mathbb{E}_{\mu_{\pi_b}} T^\pi \hat{v}(s, \theta^*)$. The gradient-based TD algorithm TDC [26] updates the parameter by minimizing the mean-square projected Bellman error (MSPBE) objective, defined as

$$J(\theta) = \mathbb{E}_{\mu_{\pi_b}}[\hat{v}(s, \theta) - \Pi T^\pi \hat{v}(s, \theta)]^2,$$

where $\Pi = \Phi(\Phi^\top \Xi \Phi)^{-1}\Phi^\top \Xi$ is the orthogonal projection operation into the function space $\hat{\mathcal{V}} = \{\hat{v}(\theta) \mid \theta \in \mathbb{R}^d \text{ and } \hat{v}(\cdot, \theta) = \phi(\cdot)^\top \theta\}$ and $\Xi$ denotes the $|\mathcal{S}| \times |\mathcal{S}|$ diagonal matrix with the components of $\mu_{\pi_b}$ as its diagonal entries. Then, we define the matrices $A$, $B$, $C$ and the vector $b$ as

$$A := \mathbb{E}_{\mu_{\pi_b}}[\rho(s,a)\phi(s)(\gamma\phi(s') - \phi(s))^\top], \quad B := -\gamma\mathbb{E}_{\mu_{\pi_b}}[\rho(s,a)\phi(s')\phi(s)^\top],$$

$$C := -\mathbb{E}_{\mu_{\pi_b}}[\phi(s)\phi(s)^\top], \quad b := \mathbb{E}_{\mu_{\pi_b}}[\rho(s,a)r(s,a,s')\phi(s)],$$

where $\rho(s,a) = \pi(a|s)/\pi_b(a|s)$ is the importance weighting factor with $\rho_{\max}$ being its maximum value. If $A$ and $C$ are both non-singular, $J(\theta)$ is strongly convex and has $\theta^* = -A^{-1}b$ as its global minimum, i.e., $J(\theta^*) = 0$. Motivated by minimizing the MSPBE objective function using the stochastic gradient methods, TDC was proposed with the following update rules:

$$\theta_{t+1} = \Pi_{R_\theta}\left(\theta_t + \alpha_t(A_t\theta_t + b_t + B_t w_t)\right), \tag{1}$$

$$w_{t+1} = \Pi_{R_w}\left(w_t + \beta_t(A_t\theta_t + b_t + C_t w_t)\right), \tag{2}$$

where $A_t = \rho(s_t, a_t)\phi(s_t)(\gamma\phi(s_{t+1}) - \phi(s_t))^\top$, $B_t = -\gamma\rho(s_t, a_t)\phi(s_{t+1})\phi(s_t)^\top$, $C_t = -\phi(s_t)\phi(s_t)^\top$, $b_t = \rho(s_t, a_t)r(s_t, a_t, s_{t+1})\phi(s_t)$, and $\Pi_R(x) = \operatorname{argmin}_{x':||x'||_2 \leq R} ||x - x'||_2$ is the projection operator onto a norm ball of radius $R < \infty$. The projection step is widely used in the stochastic approximation literature. As we will show later, iterations (1)-(2) are guaranteed to converge to the optimal parameter $\theta^*$ if we choose the value of $R_\theta$ and $R_w$ appropriately. TDC with the update rules (1)-(2) is a two time-scale algorithm. The parameter $\theta$ iterates at a slow time-scale determined by the stepsize $\{\alpha_t\}$, whereas $w$ iterates at a fast time-scale determined by the stepsize $\{\beta_t\}$. Throughout the paper, we make the following standard assumptions [3, 32, 17].

**Assumption 1** (Problem solvability). *The matrix $A$ and $C$ are non-singular.*

**Assumption 2** (Bounded feature). $||\phi(s)||_2 \leq 1$ *for all $s \in \mathcal{S}$ and $\rho_{\max} < \infty$.*

**Assumption 3** (Geometric ergodicity). *There exist constants $m > 0$ and $\rho \in (0,1)$ such that*

$$\sup_{s \in \mathcal{S}} d_{TV}(\mathbb{P}(s_t \in \cdot|s_0 = s), \mu_{\pi_b}) \leq m\rho^t, \forall t \geq 0,$$

*where $d_{TV}(P, Q)$ denotes the total-variation distance between the probability measures $P$ and $Q$.*

In Assumption 1, the matrix $A$ is required to be non-singular so that the optimal parameter $\theta^* = -A^{-1}b$ is well defined. The matrix $C$ is non-singular when the feature matrix $\Phi$ has linearly independent columns. Assumption 2 can be ensured by normalizing the basis functions $\{\phi_i\}_{i=1}^d$ and when $\pi_b(\cdot|s)$ is non-degenerate for all $s$. Assumption 3 holds for any time-homogeneous Markov chain with finite state-space and any uniformly ergodic Markov chains with general state space. Throughout the paper, we require $R_\theta \geq \|A\|_2 \|b\|_2$ and $R_w \geq 2 \left\|C^{-1}\right\|_2 \|A\|_2 R_\theta$. In practice, we can estimate $A$, $C$ and $b$ as mentioned in [3] or simply let $R_\theta$ and $R_w$ to be large enough.

## 3 Main Theorems

### 3.1 Non-asymptotic Analysis under Diminishing Stepsize

Our first main result is the convergence rate of two time-scale TDC with diminishing stepsize. We define the tracking error: $z_t = w_t - \psi(\theta_t)$, where $\psi(\theta_t) = -C^{-1}(b + A\theta_t)$ is the stationary point of the ODE given by $\dot{w}(t) = Cw(t) + A\theta_t + b$, with $\theta_t$ being fixed. Let $\lambda_\theta$ and $\lambda_w$ be any constants that satisfy $\lambda_{\max}(2A^\top C^{-1}A) \leq \lambda_\theta < 0$ and $\lambda_{\max}(2C) \leq \lambda_w < 0$.

**Theorem 1.** *Consider the projected two time-scale TDC algorithm in* (1)-(2). *Suppose Assumptions 1-3 hold. Suppose we apply diminishing stepsize* $\alpha_t = \frac{c_\alpha}{(1+t)^\sigma}$, $\beta_t = \frac{c_\beta}{(1+t)^\nu}$ *which satisfy* $0 < \nu < \sigma < 1$, $0 < c_\alpha < \frac{1}{|\lambda_\theta|}$ *and* $0 < c_\beta < \frac{1}{|\lambda_w|}$. *Suppose* $\epsilon$ *and* $\epsilon'$ *can be any constants in* $(0, \sigma - \nu]$ *and* $(0, 0.5]$, *respectively. Then we have for* $t \geq 0$:

$$\mathbb{E}\|\theta_t - \theta^*\|_2^2 \leq \mathcal{O}(e^{\frac{-|\lambda_\theta|c_\alpha}{1-\sigma}(t^{1-\sigma}-1)}) + \mathcal{O}\Big(\frac{\log t}{t^\sigma}\Big) + \mathcal{O}\Big(\frac{\log t}{t^\nu} + h(\sigma, \nu)\Big)^{1-\epsilon'}, \tag{3}$$

$$\mathbb{E}\|z_t\|_2^2 \leq \mathcal{O}\Big(\frac{\log t}{t^\nu}\Big) + \mathcal{O}(h(\sigma, \nu)), \tag{4}$$

*where*

$$h(\sigma, \nu) = \begin{cases} \frac{1}{t^\nu}, & \sigma > 1.5\nu, \\ \frac{1}{t^{2(\sigma-\nu)-\epsilon}}, & \nu < \sigma \leq 1.5\nu. \end{cases} \tag{5}$$

*If* $0 < \nu < \sigma = 1$, *with* $c_\alpha = \frac{1}{|\lambda_\theta|}$ *and* $0 < c_\beta < \frac{1}{|\lambda_w|}$, *we have for* $t \geq 0$

$$\mathbb{E}\|\theta_t - \theta^*\|_2^2 \leq \mathcal{O}\Big(\frac{(\log t)^2}{t}\Big) + \mathcal{O}\Big(\frac{\log t}{t^\nu} + h(1, \nu)\Big)^{1-\epsilon'}. \tag{6}$$

*For explicit expressions of* (3), (4) *and* (6), *please refer to* (25), (18) *and* (28) *in the Appendix.*

We further explain Theorem 1 as follows: (a) In (3) and (5), since both $\epsilon$ and $\epsilon'$ can be arbitrarily small, the convergence of $\mathbb{E}\|\theta_t - \theta^*\|_2^2$ can be almost as fast as $\frac{1}{t^{2(\sigma-\nu)}}$ when $\nu < \sigma < 1.5\nu$, and $\frac{\log t}{t^\nu}$ when $1.5\nu \leq \sigma$. Then best convergence rate is almost as fast as $\mathcal{O}(\frac{\log t}{t^{2/3}})$ with $\sigma = \frac{3}{2}\nu = 1$. (b) If data are i.i.d. generated, then our bound reduces to $\mathbb{E}\|\theta_t - \theta^*\|_2^2 \leq \mathcal{O}(\exp(\lambda_\theta c_\alpha(t^{1-\sigma} - 1)/(1 - \sigma))) + \mathcal{O}(1/t^\sigma) + \mathcal{O}(h(\sigma, \nu))^{1-\epsilon'}$ with $h(\sigma, \nu) = \frac{1}{t^\nu}$ when $\sigma > 1.5\nu$, and $h(\sigma, \nu) = \frac{1}{t^{2(\sigma-\nu)-\epsilon}}$ when $\nu < \sigma \leq 1.5\nu$. The best convergence rate is almost as fast as $\frac{1}{t^{2/3}}$ with $\sigma = \frac{3}{2}\nu = 1$ as given in [8].

Theorem 1 characterizes the relationship between the convergence rate of $\theta_t$ and stepsizes $\alpha_t$ and $\beta_t$. The first term of the bound in (3) corresponds to the convergence rate of $\theta_t$ with full gradient $\nabla J(\theta_t)$, which exponentially decays with $t$. The second term is introduced by the bias and variance of the gradient estimator which decays sublinearly with $t$. The last term arises due to the accumulated tracking error $z_t$, which specifically arises in two time-scale algorithms, and captures how accurately $w_t$ tracks $\psi(\theta_t)$. Thus, if $w_t$ tracks the stationary point $\psi(\theta_t)$ in each step perfectly, then we have only the first two terms in (3), which matches the results of one time-scale TD learning [3, 7]. Theorem 1 indicates that asymptotically, (3) is dominated by the tracking error term $\mathcal{O}(h(\sigma, \nu)^{1-\epsilon'})$, which depends on the diminishing rate of $\alpha_t$ and $\beta_t$. Since both $\epsilon$ and $\epsilon'$ can be arbitrarily small, if the diminishing rate of $\alpha_t$ is close to that of $\beta_t$, then the tracking error is dominated by the slow drift, which has an approximate order $\mathcal{O}(1/t^{2(\sigma-\nu)})$; if the diminishing rate of $\alpha_t$ is much faster than that of $\beta_t$, then the tracking error is dominated by the accumulated bias, which has an approximate order $\mathcal{O}(\log t/t^\nu)$. Moreover, (5) and (6) suggest that for any fixed $\sigma \in (0, 1]$, the optimal diminishing rate of $\beta_t$ is achieved by $\sigma = \frac{3}{2}\nu$.

From the technical standpoint, we develop novel techniques to handle the interaction between the training error and the tracking error and sharpen the error bounds recursively. The proof sketch and the detailed steps are provided in Appendix A.

## 3.2 Non-asymptotic Analysis under Constant Stepsize

As we remark in Section 1, it has been demonstrated by empirical results [9] that the standard TD under constant stepsize not only converges fast, but also has comparable training error as that under diminishing stepsize. However, this does not hold for TDC. When the two variables in TDC are updated both under constant stepsize, our experiments demonstrate that constant stepsize yields fast convergence, but has large training error. In this subsection, we aim to explain why this happens by analyzing the convergence rate of the two variables in TDC, and the impact of one on the other.

The following theorem provides the convergence result for TDC with the two variables iteratively updated respectively by two different constant stepsizes.

**Theorem 2.** *Consider the projected TDC algorithm in eqs.* (1) *and* (2)*. Suppose Assumption 1-3 hold. Suppose we apply constant stepsize $\alpha_t = \alpha$, $\beta_t = \beta$ and $\alpha = \eta\beta$ which satisfy $\eta > 0$, $0 < \alpha < \frac{1}{|\lambda_\theta|}$ and $0 < \beta < \frac{1}{|\lambda_w|}$. We then have for $t \geq 0$:*

$$\mathbb{E}\|\theta_t - \theta^*\|_2^2 \leq (1 - |\lambda_\theta|\alpha)^t (\|\theta_0 - \theta^*\|_2^2 + C_1)$$
$$+ C_2 \max\{\alpha, \alpha \ln \frac{1}{\alpha}\} + (C_3 \max\{\beta, \beta \ln \frac{1}{\beta}\} + C_4\eta)^{0.5} \qquad (7)$$

$$\mathbb{E}\|z_t\|_2^2 \leq (1 - |\lambda_w|\beta)^t \|z_0\|_2^2 + C_5 \max\{\beta, \beta \ln \frac{1}{\beta}\} + C_6\eta, \qquad (8)$$

*where $C_1 = 4\gamma\rho_{\max}R_\theta R_w \frac{1-(1-|\lambda_\theta|\alpha)^{T+1}}{|\lambda_\theta|(1-|\lambda_\theta|\alpha)^{T+1}}$ with $T = \lceil \frac{\ln[C_5 \max\{\beta, \ln(\frac{1}{\beta})\beta\}/\|z_0\|_2^2]}{-\ln(1-|\lambda_w|\beta)} \rceil$, and $C_2$, $C_3$, $C_4$, $C_5$ and $C_6$ are positive constants independent of $\alpha$ and $\beta$. For explicit expressions of $C_2$, $C_3$, $C_4$, $C_5$ and $C_6$, please refer to* (67), (68), (69), (59), *and* (60) *in the Supplementary Materials.*

Theorem 2 shows that TDC with constant stepsize converges to a neighborhood of $\theta^*$ exponentially fast. The size of the neighborhood depends on the second and the third terms of the bound in (7), which arise from the bias and variance of the update of $\theta_t$ and the tracking error $z_t$ in (8), respectively. Clearly, the convergence $z_t$, although is also exponentially fast to a neighborhood, is under a different rate due to the different condition number. We further note that as the stepsize parameters $\alpha$, $\beta$ approach 0 in a way such that $\alpha/\beta \to 0$, $\theta_t$ approaches to $\theta^*$ as $t \to \infty$, which matches the asymptotic convergence result for two time-scale TDC under constant stepsize in [36].

**Diminishing vs Constant Stepsize:** We next discuss the comparison between TDC under diminishing stepsize and constant stepsize. Generally, Theorem 1 suggests that diminishing stepsize yields better converge guarantee (i.e., converges exactly to $\theta^*$) than constant stepsize shown in Theorem 2 (i.e., converges to the neighborhood of $\theta^*$). In practice, constant stepsize is recommended because diminishing stepsize may take much longer time to converge. However, as Figure 2 in Section 4.2 shows, although TDC with large constant stepsize converges fast, the training error due to the convergence to the neighborhood is significantly worse than the diminishing stepsize. More specifically, when $\eta = \alpha/\beta$ is fixed, as $\alpha$ grows, the convergence becomes faster, but as a consequence, the term $(C_3 \max\{\beta, \beta \ln \frac{1}{\beta}\} + C_4\eta)^{0.5}$ due to the tracking error increases and results in a large training error. Alternatively, if $\alpha$ gets small so that the training error is comparable to that under diminishing stepsize, then the convergence becomes very slow. This suggests that simply setting the stepsize to be constant for TDC does not yield desired performance. This motivates us to design an appropriate update scheme for TDC such that it can enjoy as fast error convergence rate as constant stepsize offers, but still have comparable accuracy as diminishing stepsize enjoys.

## 3.3 TDC under Blockwise Diminishing Stepsize

In this subsection, we propose a blockwise diminishing stepsize scheme for TDC (see Algorithm 1), and study its theoretical convergence guarantee. In Algorithm 1, we define $t_s = \sum_{i=0}^{s} T_s$.

The idea of Algorithm 1 is to divide the iteration process into blocks, and diminish the stepsize blockwisely, but keep the stepsize to be constant within each block. In this way, within each block,

---

**Algorithm 1** Blockwise Diminishing Stepsize TDC

---

**Input:** $\theta_{0,0} = \theta_0$, $w_{0,0} = w_0 = 0$, $T_0 = 0$, block index $S$
 1: **for** $s = 1, 2, ..., S$ **do**
 2:   $\theta_{s,0} = \theta_{s-1}$, $w_{s,0} = w_{s-1}$
 3:   **for** $i = 1, 2, ..., T_s$ **do**
 4:     Sample $(s_{t_{s-1}+i}, a_{t_{s-1}+i}, s_{t_{s-1}+i+1}, r_{t_{s-1}+i})$ from trajectory
 5:     $\theta_{s,i} = \Pi_{R_\theta}\left(\theta_{s,i-1} + \alpha_s(A_{t_{s-1}+i}\theta_{s,i-1} + b_{t_{s-1}+i} + B_{t_{s-1}+i}w_{s,i-1})\right)$
 6:     $w_{s,i} = \Pi_{R_w}\left(w_{s,i-1} + \beta_s(A_{t_{s-1}+i}\theta_{s,i-1} + b_{t_{s-1}+i} + C_{t_{s-1}+i}w_{s,i-1})\right)$
 7:   **end for**
 8:   $\theta_s = \theta_{s,T_s}$, $w_s = w_{s,T_s}$
 9: **end for**
**Output:** $\theta_S, w_S$

---

TDC can decay fast due to constant stepsize and still achieve an accurate solution due to blockwisely decay of the stepsize, as we will demonstrate in Section 4. More specifically, the constant stepsizes $\alpha_s$ and $\beta_s$ for block $s$ are chosen to decay geometrically, such that the tracking error and accumulated variance and bias are asymptotically small; and the block length $T_s$ increases geometrically across blocks, such that the training error $\mathbb{E}\|\theta_s - \theta^*\|_2^2$ decreases geometrically blockwisely. We note that the design of the algorithm is inspired by the method proposed in [35] for conventional optimization problems.

The following theorem characterizes the convergence of Algorithm 1.

**Theorem 3.** *Consider the projected TDC algorithm with blockwise diminishing stepsize as in Algorithm 1. Suppose Assumptions 1-3 hold. Suppose $\max\{\log(1/\alpha_s)\alpha_s, \alpha_s\} \leq \min\{\epsilon_{s-1}/(4C_7), 1/|\lambda_x|\}$, $\beta_s = \eta\alpha_s$ and $T_s = \lceil\log_{1/(1-|\lambda_x|\alpha_s)} 4\rceil$, where $\lambda_x < 0$ and $C_7 > 0$ are constant independent of $s$ (see (72) and (75) in the Supplementary Materials for explicit expression of $\lambda_x$ and $C_7$), $\epsilon_s = \|\theta_0 - \theta^*\|_2 /2^s$ and $\eta \geq 1/2\max\{0, \lambda_{\min}(C^{-1}(A^\top + A))\}$. Then, after $S = \lceil\log_2(\epsilon_0/\epsilon)\rceil$ blocks, we have*

$$\mathbb{E}\|\theta_S - \theta^*\|_2^2 \leq \epsilon.$$

*The total sample complexity is $\mathcal{O}(\frac{1}{\epsilon}\log\frac{1}{\epsilon})$.*

Theorem 3 indicates that the sample complexity of TDC under blockwise diminishing stepsize is slightly better than that under diminishing stepsize. Our empirical results (see Section 4.3) also demonstrate that blockwise diminishing stepsize yields as fast convergence as constant stepsize and has comparable training error as diminishing stepsize. However, we want to point out that the advantage of blockwise diminishing stepsize does not come for free, rather at the cost of some extra parameter tuning in practice to estimate $\epsilon_0$, $|\lambda_x|$, $C_7$ and $\eta$; whereas diminishing stepsize scheme as guided by our Theorem 1 requires to tune at most three parameters to obtain desirable performance.

## 4 Experimental Results

In this section, we provide numerical experiments to verify our theoretical results and the efficiency of Algorithm 1. More precisely, we consider Garnet problems [1] denoted as $\mathcal{G}(n_S, n_A, p, q)$, where $n_s$ denotes the number of states, $n_A$ denotes the number of actions, $p$ denotes the number of possible next states for each state-action pair, and $q$ denotes the number of features. The reward is state-dependent and both the reward and the feature vectors are generated randomly. The discount factor $\gamma$ is set to 0.95 in all experiments. We consider the $\mathcal{G}(500, 20, 50, 20)$ problem. For all experiments, we choose $\theta_0 = w_0 = 0$. All plots report the evolution of the mean square error over 500 independent runs.

### 4.1 Optimal Diminishing Stepsize

In this subsection, we provide numerical results to verify Theorem 1. We compare the performance of TDC updates with the same $\alpha_t$ but different $\beta_t$. We consider four different diminishing stepsize settings: (1) $c_\alpha = c_\beta = 0.03$, $\sigma = 0.15$; (2) $c_\alpha = c_\beta = 0.18$, $\sigma = 0.30$; (3) $c_\alpha = c_\beta = 1$, $\sigma = 0.45$; (4) $c_\alpha = c_\beta = 4$, $\sigma = 0.60$. For each case with fixed slow time-scale parameter $\sigma$, the fast time-scale stepsize $\beta_t$ has decay rate $\nu$ to be $\frac{1}{2}\sigma$, $\frac{1}{3}\sigma$, $\frac{5}{9}\sigma$, $\frac{2}{3}\sigma$, $\frac{5}{6}\sigma$, and $\sigma$. Our results are reported in Figure 1,

in which for each case the left figure reports the overall iteration process and the right figure reports the corresponding zoomed tail process of the last 100000 iterations. It can be seen that in all cases, TDC iterations with the same slow time-scale stepsize $\sigma$ share similar error decay rates (see the left plot), and the difference among the fast time-scale parameter $\nu$ is reflected by the behavior of the error convergence tails (see the right plot). We observe that $\nu = \frac{2}{3}\sigma$ yields the best error decay rate. This corroborates Theorem 1, which illustrates that the fast time-scale stepsize $\beta_t$ with parameter $\nu$ affects only the tracking error term in (3), that dominates the error decay rate asymptotically.

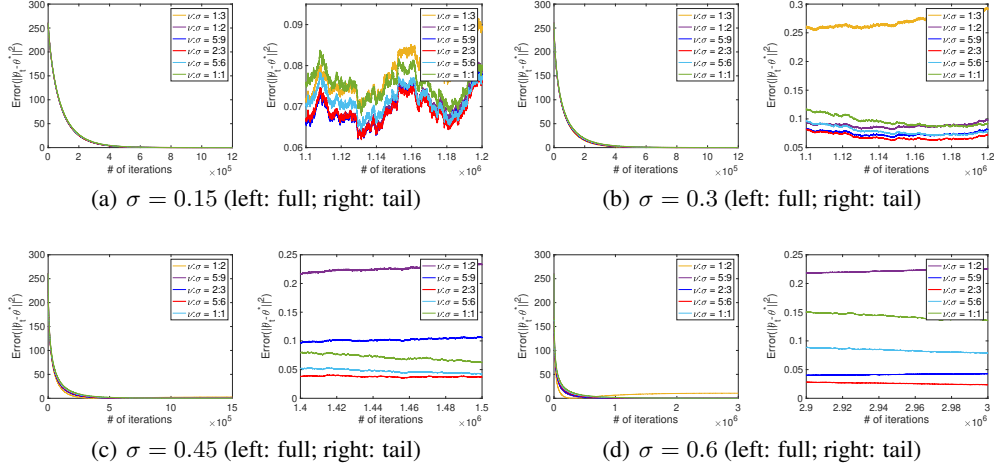

(a) $\sigma = 0.15$ (left: full; right: tail)          (b) $\sigma = 0.3$ (left: full; right: tail)

(c) $\sigma = 0.45$ (left: full; right: tail)          (d) $\sigma = 0.6$ (left: full; right: tail)

Figure 1: Comparison among diminishing stepsize settings. For settings $\sigma = 0.45$ and $\sigma = 0.6$, the case $\nu : \sigma = 1 : 3$ has much larger training error than others and is not included in the tail figures.

## 4.2 Constant Stepsize vs Diminishing Stepsize

In this subsection, we compare the error decay of TDC under diminishing stepsize with that of TDC under four different constant stepsizes. For diminishing stepsize, we set $c_\alpha = c_\beta$ and $\sigma = \frac{3}{2}\nu$, and tune their values to the best, which are given by $c_\alpha = c_\beta = 1.8$, $\sigma = \frac{3}{2}\nu = 0.45$. For the four constant-stepsize cases, we fix $\alpha$ for each case, and tune $\beta$ to the best. The resulting parameter settings are respectively as follows: $\alpha_t = 0.01$, $\beta_t = 0.006$; $\alpha_t = 0.02$, $\beta_t = 0.008$; $\alpha_t = 0.05$, $\beta_t = 0.02$; and $\alpha_t = 0.1$, $\beta_t = 0.02$. The results are reported in Figure 2, in which for both the training and tracking errors, the left plot illustrates the overall iteration process and the right plot illustrates the corresponding zoomed error tails. The results suggest that although some large constant stepsizes ($\alpha_t = 0.05$, $\beta_t = 0.02$ and $\alpha_t = 0.1$, $\beta_t = 0.02$) yield initially faster convergence than diminishing stepsize, they eventually oscillate around a large neighborhood of $\theta^*$ due to the large tracking error. Small constant stepsize ($\alpha_t = 0.02$, $\beta_t = 0.008$ and $\alpha_t = 0.01$, $\beta_t = 0.006$) can have almost the same asymptotic accuracy as that under diminishing stepsize, but has very slow convergence rate. We can also observe strong correlation between the training and tracking errors under constant stepsize, i.e., larger training error corresponds to larger tracking error, which corroborates Theorem 2 and suggests that the accuracy of TDC heavily depends on the decay of the tracking error $\|z_t\|_2$.

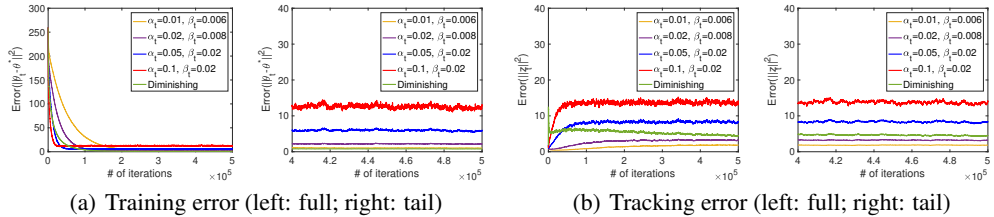

(a) Training error (left: full; right: tail)          (b) Tracking error (left: full; right: tail)

Figure 2: Comparison between TDC updates under constant stepsizes and diminishing stepsize.

### 4.3 Blockwise Diminishing Stepsize

In this subsection, we compare the error decay of TDC under blockwise diminishing stepsize with that of TDC under diminishing stepsize and constant stepsize. We use the best tuned parameter settings as listed in Section 4.2 for the latter two algorithms, i.e., $c_\alpha = c_\beta = 1.8$ and $\sigma = \frac{3}{2}\nu = 0.45$ for diminishing stepsize, and $\alpha_t = 0.1$, $\beta_t = 0.02$ for constant stepsize. We report our results in Figure 3. It can be seen that TDC under blockwise diminishing stepsize converges faster than that under diminishing stepsize and almost as fast as that under constant stepsize. Furthermore, TDC under blockwise diminishing stepsize also has comparable training error as that under diminishing stepsize. Since the stepsize decreases geometrically blockwisely, the algorithm approaches to a very small neighborhood of $\theta^*$ in the later blocks. We can also observe that the tracking error under blockwise diminishing stepsize decreases rapidly blockwisely.

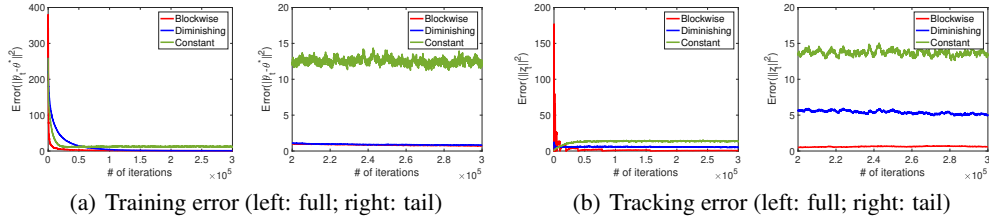

(a) Training error (left: full; right: tail)    (b) Tracking error (left: full; right: tail)

Figure 3: Comparison between TDC updates under blockwise diminishing stepsizes, diminishing stepsize and constant stepsize

### 4.4 Robustness to Blocksize

In this subsection, we investigate the robustness of TDC under blockwise diminishing stepsize with respect to the blocksize. We consider the same setting as in Section 4.3, and perturb all blocksizes by certain percentages of the original blocksize suggested in the algorithm. It can be seen from Figure 4 that the error decay rate changes only very slightly even with a substantial change in the blocksize.

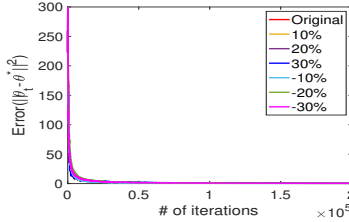

Figure 4: Comparison between TDC updates under blockwise diminishing stepsizes with different blocksizes.

## 5 Conclusion

In this work, we provided the first non-asymptotic analysis for the two time-scale TDC algorithm over Markovian sample path. We developed a novel technique to handle the accumulative tracking error caused by the two time-scale update, using which we characterized the non-asymptotic convergence rate with general diminishing stepsize and constant stepsize. We also proposed a blockwise diminishing stepsize scheme for TDC and proved its convergence. Our experiments demonstrated the performance advantage of such an algorithm over both the diminishing and constant stepsize TDC algorithms. Our technique for non-asymptotic analysis of two time-scale algorithms can be applied to studying other off-policy algorithms such as actor-critic [18] and gradient Q-learning algorithms [19].

## Acknowledgment

The work of T. Xu and Y. Liang was supported in part by the U.S. National Science Foundation under the grants CCF-1761506, ECCS-1818904, and CCF-1801855.

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
