[Supplementary Material · two-time-scale-supp.pdf]

# Supplementary Materials

## A  Technical Proofs for TDC under Decreasing Stepsize

We present the proof of Theorem 1 in four subsections. Section A.1 provides the proof sketch. Section A.2 contains the main part of the proof. Section A.3 includes all technical lemmas for the convergence proof of fast time-scale iteration, and Section A.4 includes all the technical lemmas for the convergence proof of the slow time-scale iteration.

### A.1  Proof Sketch of Theorem 1

*Proof Sketch of Theorem 1.* The proof consists of four steps as we briefly describe here. The details are provided in Appendix A.2.

**Step 1.** *Formulate training and tracking error updates.* In stead of investigating the convergence of $\{\theta_t\}$ and $\{w_t\}$ directly, we substitute $z_t$ into the TDC update (1)-(2) and analyze the update of TDC in terms of $\{\theta_t\}$ and tracking error $\{z_t\}$.

**Step 2.** *Derive preliminary bound on $\mathbb{E}\|z_t\|_2^2$.* We decompose the mean square tracking error $\mathbb{E}\|z_t\|_2^2$ into an exponentially decaying term, a variance term, a bias term, and a slow drift term, and bound each term individually. We obtain a preliminary upper bound on $\mathbb{E}\|z_t\|_2^2$ with order $\mathcal{O}(1/t^{\sigma-\nu})$.

**Step 3.** *Recursively refine bound on $\mathbb{E}\|z_t\|_2^2$.* By recursively substituting the preliminary bound of $\mathbb{E}\|z_t\|_2^2$ into the slow drift term, we obtain the refined decay rate $\mathbb{E}\|z_t\|_2^2 = \mathcal{O}(h(\sigma, \nu))$.

**Step 4.** *Derive bound on $\mathbb{E}\|\theta_t - \theta^*\|_2^2$.* We decompose the training error $\mathbb{E}\|\theta_t - \theta^*\|_2^2$ into an exponentially decaying term, a variance term, a bias term, and a tracking error term, and bound each term individually. We then recursively substitute the decay rate of $\mathbb{E}\|z_t\|_2^2$ and $\mathbb{E}\|\theta_t - \theta^*\|_2^2$ into the tracking error term to obtain an upper bound on the training error with order $\mathcal{O}(h(\sigma, \nu)^{1-\epsilon'})$. Combining each term yields the final bound of $\mathbb{E}\|\theta_t - \theta^*\|_2^2$ in (3). □

### A.2  Proof of Theorem 1

We provide the proof of Theorem 1 following four steps.

**Step 1.** *Formulation of training error and tracking error update.* We define the tracking error vector $z_t = w_t + C^{-1}(b + A\theta_t)$. By substituting $z_t$ into (1)-(2), we can rewrite the update rule of TDC in terms of $\theta_t$ and $z_t$ as follows:

$$\theta_{t+1} = \Pi_{R_\theta}\left(\theta_t + \alpha_t(f_1(\theta_t, O_t) + g_1(z_t, O_t))\right), \tag{9}$$

$$z_{t+1} = \Pi_{R_w}\left(z_t + \beta_t(f_2(\theta_t, O_t) + g_2(z_t, O_t)) - C^{-1}(b + A\theta_t)\right) + C^{-1}(b + A\theta_{t+1}), \tag{10}$$

where

$$f_1(\theta_t, O_t) = (A_t - B_t C^{-1} A)\theta_t + (b_t - B_t C^{-1} b), \qquad g_1(z_t, O_t) = B_t z_t,$$
$$f_2(\theta_t, O_t) = (A_t - C_t C^{-1} A)\theta_t + (b_t - C_t C^{-1} b), \qquad g_2(z_t, O_t) = C_t z_t,$$

with $O_t = (s_t, a_t, r_t, s_{t+1})$ denoting the observation at time step $t$. We further define

$$\bar{f}_1(\theta_t) = (A - BC^{-1}A)\theta_t + (b - BC^{-1}b), \qquad \bar{g}_1(z_t) = Bz_t, \qquad \bar{g}_2(z_t) = Cz_t.$$

**Step 2.** *Derive preliminary bound on* $\mathbb{E}\|z_t\|_2^2$. We bound the recursion of the tracking error vector $z_t$ in (10) as follows. For any $t \geq 0$, we derive

$$
\begin{aligned}
\|z_{t+1}\|_2^2 &= \left\|\Pi_{R_w}\left(z_t + \beta_t(f_2(\theta_t, O_t) + g_2(z_t, O_t)) - C^{-1}(b + A\theta_t)\right) + C^{-1}(b + A\theta_{t+1})\right\|_2^2 \\
&= \left\|\Pi_{R_w}\left(z_t + \beta_t(f_2(\theta_t, O_t) + g_2(z_t, O_t)) - C^{-1}(b + A\theta_t)\right) + \Pi_{R_w}\left(C^{-1}(b + A\theta_{t+1})\right)\right\|_2^2 \\
&\leq \left\|z_t + \beta_t(f_2(\theta_t, O_t) + g_2(z_t, O_t)) + C^{-1}A(\theta_{t+1} - \theta_t)\right\|_2^2 \\
&= \|z_t\|_2^2 + 2\beta_t\langle f_2(\theta_t, O_t), z_t\rangle + 2\beta_t\langle g_2(z_t, O_t), z_t\rangle + 2\langle C^{-1}A(\theta_{t+1} - \theta_t), z_t\rangle \\
&\quad + \left\|\beta_t f_2(\theta_t, O_t) + \beta_t g_2(z_t, O_t) + C^{-1}A(\theta_{t+1} - \theta_t)\right\|_2^2 \\
&\leq \|z_t\|_2^2 + 2\beta_t\langle \bar{g}_2(z_t), z_t\rangle + 2\beta_t\langle f_2(\theta_t, O_t), z_t\rangle + 2\beta_t\langle g_2(z_t, O_t) - \bar{g}_2(z_t), z_t\rangle \\
&\quad + 2\langle C^{-1}A(\theta_{t+1} - \theta_t), z_t\rangle \\
&\quad + 3\beta_t^2\|f_2(\theta_t, O_t)\|_2^2 + 3\beta_t^2\|g_2(z_t, O_t)\|_2^2 + 3\left\|C^{-1}A(\theta_{t+1} - \theta_t)\right\|_2^2 \\
&\leq \|z_t\|_2^2 + 2\beta_t\langle Cz_t, z_t\rangle + 2\beta_t\langle f_2(\theta_t, O_t), z_t\rangle + 2\beta_t\langle g_2(z_t, O_t) - \bar{g}_2(z_t), z_t\rangle \\
&\quad + 2\langle C^{-1}A(\theta_{t+1} - \theta_t), z_t\rangle + 3\beta_t^2\|f_2(\theta_t, O_t)\|_2^2 + 3\beta_t^2\|g_2(z_t, O_t)\|_2^2 \\
&\quad + 3\alpha_t^2\left\|C^{-1}\right\|_2^2\|A\|_2^2\|f_1(\theta_t, O_t) + g_1(z_t, O_t)\|_2^2 \\
&\leq (1 - \beta_t|\lambda_w|)\|z_t\|_2^2 + 2\beta_t\zeta_{f_2}(\theta_t, z_t, O_t) + 2\beta_t\zeta_{g_2}(z_t, O_t) + 2\langle C^{-1}A(\theta_{t+1} - \theta_t), z_t\rangle \\
&\quad + 3\beta_t^2 K_{f_2}^2 + 3\beta_t^2 K_{g_2}^2 + 6\alpha_t^2\left\|C^{-1}\right\|_2^2\|A\|_2^2(K_{f_1}^2 + K_{g_1}^2),
\end{aligned}
$$

where $\lambda_{\max}(2C) \leq \lambda_w < 0$, $\zeta_{f_2}(\theta_t, z_t, O_t) = \langle f_2(\theta_t, O_t), z_t\rangle$, $\zeta_{g_2}(z_t, O_t) = \langle g_2(z_t, O_t) - \bar{g}_2(z_t), z_t\rangle$. $K_{f_1}$ and $K_{g_1}$, $K_{f_2}$ and $K_{g_1}$ are positive constants, please refer to Lemma 12, 13, 2 and 6 for their definitions. Then, defining $K_{r_1} = \left\|C^{-1}\right\|_2\|A\|_2(K_{f_1} + K_{g_1})$ and taking the expectation over $\mathcal{F}_{t+1}$ (the filtration up to state $s_{t+1}$) on both sides, we have

$$
\begin{aligned}
\mathbb{E}\|z_{t+1}\|_2^2 &\leq (1 - \beta_t|\lambda_w|)\mathbb{E}\|z_t\|_2^2 + 2\beta_t\mathbb{E}[\zeta_{f_2}(\theta_t, z_t, O_t)] + 2\beta_t\mathbb{E}[\zeta_{g_2}(z_t, O_t)] \\
&\quad + 2\mathbb{E}\langle C^{-1}A(\theta_{t+1} - \theta_t), z_t\rangle + 3\beta_t^2 K_{f_2}^2 + 3\beta_t^2 K_{g_2}^2 + 6\alpha_t^2 K_{r_1}^2.
\end{aligned} \tag{11}
$$

From the definition of $\beta_t$, $|\beta_t| \leq c_\beta$ for all $t \geq 0$. If $c_\beta|\lambda_w| < 1$, then we have $0 < 1 - \beta_t|\lambda_w| < 1$. Telescoping the above inequality yields that

$$
\begin{aligned}
\mathbb{E}\|z_{t+1}\|_2^2 &\leq \left[\prod_{i=0}^{t}(1 - \beta_i|\lambda_w|)\right]\|z_0\|_2^2 \\
&\quad + 2\sum_{i=0}^{t}\left[\prod_{k=i+1}^{t}(1 - \beta_k|\lambda_w|)\right]\beta_i[\zeta_{f_2}(\theta_i, z_i, O_i)] \\
&\quad + 2\sum_{i=0}^{t}\left[\prod_{k=i+1}^{t}(1 - \beta_k|\lambda_w|)\right]\beta_i[\zeta_{g_2}(z_i, O_i)] \\
&\quad + 2\sum_{i=0}^{t}\left[\prod_{k=i+1}^{t}(1 - \beta_k|\lambda_w|)\right]\mathbb{E}\langle C^{-1}A(\theta_{i+1} - \theta_i), z_i\rangle \\
&\quad + 3(K_{f_2}^2 + K_{g_2}^2)\sum_{i=0}^{t}\left[\prod_{k=i+1}^{t}(1 - \beta_k|\lambda_w|)\right]\beta_i^2 + 6K_{r_1}^2\sum_{i=0}^{t}\left[\prod_{k=i+1}^{t}(1 - \beta_k|\lambda_w|)\right]\alpha_i^2.
\end{aligned} \tag{12}
$$

Since $1 - \beta_i|\lambda_w| \leq e^{-\beta_i|\lambda_w|}$ and using the fact that $(1+i)^{-\nu} \geq (1+i)^{-\sigma}$ for all $i \geq 0$, we have

$$\mathbb{E}\left\|z_{t+1}\right\|_2^2 \leq e^{-|\lambda_w|\sum_{i=0}^t \beta_i} \left\|z_0\right\|_2^2 \tag{13}$$

$$+ 2\sum_{i=0}^t e^{-|\lambda_w|\sum_{k=i+1}^t \beta_k} \beta_i \mathbb{E}[\zeta_{f_2}(\theta_i, z_i, O_i)] \tag{14}$$

$$+ 2\sum_{i=0}^t e^{-|\lambda_w|\sum_{k=i+1}^t \beta_k} \beta_i \mathbb{E}[\zeta_{g_2}(z_i, O_i)] \tag{15}$$

$$+ 2\sum_{i=0}^t e^{-|\lambda_w|\sum_{k=i+1}^t \beta_k} \mathbb{E}\langle C^{-1}A(\theta_{i+1} - \theta_i), z_i\rangle \tag{16}$$

$$+ 3\max\{1, \frac{c_\alpha^2}{c_\beta^2}\}(K_{f_2}^2 + K_{g_2}^2 + 3K_{r_1}^2)\sum_{i=0}^t e^{-|\lambda_w|\sum_{k=i+1}^t \beta_k} \beta_i^2. \tag{17}$$

The first term (13) captures how fast the tracking error vector $z_t$ converges to the neighborhood of zero, the second term (14) and third term (15) are the accumulative bias term induced by the biased gradient estimator $f_2(\theta_t, O_t)$ and $g_2(z_t, O_t)$ respectively, the forth term (16) is the accumulative error caused by the slow drift, and the last term (17) is the accumulative variance. Combining Lemma 5, Lemma 9, Lemma 10 and applying Lemma 1 to the above upper bound, we obtain

$$\mathbb{E}\left\|z_{t+1}\right\|_2^2$$
$$\leq e^{\frac{-|\lambda_w|c_\beta}{1-\nu}[(1+t)^{1-\nu}-1]}\left\|z_0\right\|_2^2 + 8(R_w K_{f_2} + 2R_w K_{g_2})\frac{e^{|\lambda_w|c_\beta}}{|\lambda_w|}\frac{c_\beta}{(1+t)^\nu}$$

$$+ 2c_\beta[K_{r_3} + L_{g_2,z}K_{r_2}]\tau_\beta\frac{e^{|\lambda_w|c_\beta}}{|\lambda_w|}e^{\frac{-|\lambda_w|c_\beta}{1-\nu}[(1+t)^\nu-(1+\tau_\beta)^\nu]}$$

$$+ 4(K_{r_3} + L_{g_2,z}K_{r_2})\tau_\beta\frac{e^{|\lambda_w|c_\beta/2}}{|\lambda_w|}(e^{\frac{-|\lambda_w|c_\beta}{2(1-\nu)}[(t+1)^{1-\nu}-1]}D_1\mathbb{1}_{\{\tau_\beta+1<i_{d_1}\}} + \beta_{t-\tau_\beta})$$

$$+ \frac{4c_\alpha(1+\gamma)\rho_{\max}}{c_\beta\lambda_{cm}}R_\theta R_w\frac{2e^{|\lambda_w|c_\beta/2}}{|\lambda_w|}\left(e^{\frac{-|\lambda_w|c_\beta}{2(1-\nu)}[(1+t)^{1-\nu}-1]}D_2 + \frac{1}{(1+t)^{\sigma-\nu}}\right)$$

$$+ 3\max\left\{1, \frac{c_\alpha^2}{c_\beta^2}\right\}(K_{f_2}^2 + K_{g_2}^2 + K_{r_1}^2)\frac{2c_\beta e^{|\lambda_w|c_\beta/2}}{|\lambda_w|}(e^{\frac{-|\lambda_w|c_\beta}{2(1-\nu)}[(t+1)^{1-\nu}-1]}D_3 + \beta_t) \tag{18}$$

Where, $D_3 = e^{(|\lambda_w|c_\beta/2)\sum_{k=0}^{i_{d_3}}\beta_k}$, with $i_{d_3} = (\frac{|\lambda_w|c_\beta}{2\nu})^{\frac{1}{1-\nu}}$, and $\tau_\beta = \min\{i \in \mathbb{N}|m\rho^i \leq \beta_t\}$.

**Step 3.** *Recursively refine bound on* $\mathbb{E}\left\|z_t\right\|_2^2$. By applying Lemma 11, we have

$$\mathbb{E}\left\|z_t\right\|_2^2 \leq \mathcal{O}\left(\frac{\log t}{t^\nu}\right) + \mathcal{O}(h(\sigma, \nu)),$$

where

$$h(\sigma, \nu) = \begin{cases} \frac{1}{t^\nu}, & \sigma > 1.5\nu, \\ \frac{1}{t^{2(\sigma-\nu)-\epsilon}}. & \nu < \sigma \leq 1.5\nu, \end{cases}$$

where $\epsilon \in (0, \sigma - \nu]$ can be any small constant.

**Step 4.** *Derive bound on* $\mathbb{E} \|\theta_t - \theta^*\|_2^2$. For the recursion of $\theta_t$ in (9), for any $t \geq 0$,

$$
\begin{aligned}
\|\theta_{t+1} - \theta^*\|_2^2 &= \|\Pi_{R_\theta} \left(\theta_t + \alpha_t(f_1(\theta_t, O_t) + g_1(z_t, O_t))\right) - \theta^*\|_2^2 \\
&= \|\Pi_{R_\theta} \left(\theta_t + \alpha_t(f_1(\theta_t, O_t) + g_1(z_t, O_t))\right) - \Pi_{R_\theta} \theta^*\|_2^2 \\
&\leq \|\theta_t - \theta^* + \alpha_t(f_1(\theta_t, O_t) + g_1(z_t, O_t))\|_2^2 \\
&= \|\theta_t - \theta^*\|_2^2 + 2\alpha_t \langle f_1(\theta_t, O_t), \theta_t - \theta^* \rangle + 2\alpha_t \langle g_1(z_t, O_t), \theta_t - \theta^* \rangle \\
&\quad + \alpha_t^2 \|f_1(\theta_t, O_t) + g_1(z_t, O_t)\|_2^2 \\
&\leq \|\theta_t - \theta^*\|_2^2 + 2\alpha_t \langle \bar{f}_1(\theta_t), \theta_t - \theta^* \rangle + 2\alpha_t \langle f_1(\theta_t, O_t) - \bar{f}_1(\theta_t), \theta_t - \theta^* \rangle \\
&\quad + 2\alpha_t \langle g_1(z_t, O_t), \theta_t - \theta^* \rangle + 2\alpha_t^2 \|f_1(\theta_t, O_t)\|_2^2 + 2\alpha_t^2 \|g_1(z_t, O_t)\|_2^2 \\
&\leq \|\theta_t - \theta^*\|_2^2 + 2\alpha_t \langle (A^\top C^{-1} A)(\theta_t - \theta^*), \theta_t - \theta^* \rangle + 2\alpha_t \langle f_1(\theta_t, O_t) - \bar{f}_1(\theta_t), \theta_t - \theta^* \rangle \\
&\quad + 2\alpha_t \langle g_1(z_t, O_t), \theta_t - \theta^* \rangle + 2\alpha_t^2 \|f_1(\theta_t, O_t)\|_2^2 + 2\alpha_t^2 \|g_1(z_t, O_t)\|_2^2 \\
&\leq (1 - \alpha_t |\lambda_\theta|) \|\theta_t - \theta^*\|_2^2 + 2\alpha_t \zeta_{f_1}(\theta_t, O_t) + 2\alpha_t \langle B_t z_t, \theta_t - \theta^* \rangle \qquad (19) \\
&\quad + 2\alpha_t^2 \|f_1(\theta_t, O_t)\|_2^2 + 2\alpha_t^2 \|g_1(z_t, O_t)\|_2^2 ,
\end{aligned}
$$

where $2\lambda_{\max}(A^\top C^{-1} A) \leq \lambda_\theta < 0$ and $\zeta_{f_1}(\theta_t, O_t) = \langle f_1(\theta_t, O_t) - \bar{f}_1(\theta_t), \theta_t - \theta^* \rangle$.

First consider the case when $0 < \nu < \sigma < 1$. Telescoping the above inequality and taking the expectation over $\mathcal{F}_{t+1}$ on both sides yield that

$$
\begin{aligned}
\mathbb{E} \|\theta_{t+1} - \theta^*\|_2^2 &\leq \left[ \prod_{i=0}^t (1 - \alpha_i |\lambda_\theta|) \right] \|\theta_0 - \theta^*\|_2^2 \\
&\quad + 2 \sum_{i=0}^t \left[ \prod_{k=i+1}^t (1 - \alpha_i |\lambda_\theta|) \right] \alpha_i \mathbb{E} \zeta_{f_1}(\theta_i, O_i) \\
&\quad + 2 \sum_{i=0}^t \left[ \prod_{k=i+1}^t (1 - \alpha_i |\lambda_\theta|) \right] \alpha_i \mathbb{E} \langle B_i z_i, \theta_i - \theta^* \rangle \\
&\quad + 2(K_{f_1}^2 + K_{g_1}^2) \sum_{i=0}^t \left[ \prod_{k=i+1}^t (1 - \alpha_i |\lambda_\theta|) \right] \alpha_i^2. \qquad (20)
\end{aligned}
$$

Then following steps that are similar to (14)-(17), we obtain

$$
\mathbb{E} \|\theta_{t+1} - \theta^*\|_2^2 \leq e^{-|\lambda_\theta| \sum_{i=0}^t \alpha_i} \|\theta_0 - \theta^*\|_2^2 \qquad (21)
$$

$$
+ 2 \sum_{i=0}^t e^{-|\lambda_\theta| \sum_{k=i+1}^t \alpha_k} \alpha_i \mathbb{E}[\zeta_{f_1}(\theta_i, O_i)] \qquad (22)
$$

$$
+ 2 \sum_{i=0}^t e^{-|\lambda_\theta| \sum_{k=i+1}^t \alpha_k} \alpha_i \mathbb{E}[\langle B_i z_i, \theta_i - \theta^* \rangle] \qquad (23)
$$

$$
+ 2(K_{f_1}^2 + K_{g_1}^2) \sum_{i=0}^t e^{-|\lambda_\theta| \sum_{k=i+1}^t \alpha_k} \alpha_i^2. \qquad (24)
$$

Similarly, the first term (21) captures how fast $\theta_t$ converges to the neighborhood of $\theta^*$, the second term (22) represents the accumulative bias induced by the biased gradient estimator $f_1(\theta_t, O_t)$, the third term (23) represents the accumulative error caused by imperfect tracking of $w_t$, and the last term (24) captures the accumulative variance. Combining Lemma 16, Lemma 18 and applying Lemma 1

to the above upper bound, we obtain

$$\mathbb{E}\left\|\theta_{t+1}-\theta^*\right\|_2^2 \le e^{\frac{-|\lambda_\theta|c_\alpha}{1-\sigma}[(1+t)^{1-\sigma}-1]}\left\|\theta_0-\theta^*\right\|_2^2$$

$$+2c_\alpha L_{f_1,\theta}(K_{f_1}+K_{g_1})\tau_\alpha\frac{e^{|\lambda_\theta|c_\alpha}}{|\lambda_\theta|}e^{\frac{-|\lambda_\theta|c_\alpha}{1-\sigma}[(1+t)^\sigma-(1+\tau_\sigma)^\sigma]}+16R_\theta K_{f_1}\frac{e^{|\lambda_\theta|c_\alpha}}{|\lambda_\theta|}\frac{c_\alpha}{(1+t)^\sigma}$$

$$+2L_{f_1,\theta}(K_{f_1}+K_{g_1})\tau_\alpha\frac{2e^{|\lambda_\theta|c_\alpha/2}}{|\lambda_\theta|}(e^{\frac{-|\lambda_\theta|c_\alpha}{2(1-\sigma)}[(t+1)^{1-\sigma}-1]}D_4\mathbb{1}_{\{\tau_\alpha+1<i_\alpha\}}+\alpha_{t-\tau_\alpha})$$

$$+2(K_{f_1}^2+K_{g_1}^2)\frac{2c_\alpha e^{|\lambda_\theta|c_\alpha/2}}{|\lambda_\theta|}(e^{\frac{-|\lambda_\theta|c_\alpha}{2(1-\sigma)}[(t+1)^{1-\sigma}-1]}D_4+\alpha_t)$$

$$+\mathcal{O}(\frac{\log t}{t^\nu}+h(\sigma,\nu))^{1-\epsilon'}, \tag{25}$$

where $\epsilon' \in (0,0.5]$ can be any small constant, $D_4=e^{(|\lambda_\theta|c_\alpha/2)\sum_{k=0}^{i_{d_4}}\alpha_k}$, $i_{d_4}=(\frac{|\lambda_\theta|c_\alpha}{2\sigma})^{\frac{1}{1-\sigma}}$ and $\tau_\alpha=\min\{i\in\mathbb{N}|m\rho^i\le\alpha_t\}$.

If $\sigma=1$, choosing the stepsize $\alpha_t=\frac{1}{|\lambda_\theta|(1+t)}$, starting from (19) and applying Lemma 12 and Lemma 13, we have

$$\left\|\theta_{t+1}-\theta^*\right\|_2^2\le(1-\frac{1}{1+t})\left\|\theta_t-\theta^*\right\|_2^2+\frac{2}{|\lambda_\theta|(1+t)}\zeta_{f_1}(\theta_t,O_t)+\frac{2}{|\lambda_\theta|(1+t)}\langle B_t z_t,\theta_t-\theta^*\rangle$$

$$+\frac{2}{\lambda_\theta^2(1+t)^2}(K_{f_1}^2+K_{g_1}^2),$$

which further implies that

$$(1+t)\left\|\theta_{t+1}-\theta^*\right\|_2^2-t\left\|\theta_t-\theta^*\right\|_2^2$$

$$\le\frac{2}{|\lambda_\theta|}\zeta_{f_1}(\theta_t,O_t)+\frac{2}{|\lambda_\theta|}\langle B_t z_t,\theta_t-\theta^*\rangle+\frac{2(K_{f_1}^2+K_{g_1}^2)}{\lambda_\theta^2}\frac{1}{1+t}. \tag{26}$$

Applying (26) recursively and taking the expectation over $\mathcal{F}_{t+1}$ on both sides yields that

$$\mathbb{E}\left\|\theta_{t+1}-\theta^*\right\|_2^2$$

$$\le\frac{2}{|\lambda_\theta|(1+t)}\sum_{i=0}^t\mathbb{E}\zeta_{f_1}(\theta_i,O_i)+\frac{2}{|\lambda_\theta|(1+t)}\sum_{i=0}^t\mathbb{E}\langle B_i z_i,\theta_i-\theta^*\rangle+\frac{2(K_{f_1}^2+K_{g_1}^2)}{\lambda_\theta^2(1+t)}\sum_{i=0}^t\frac{1}{1+i}. \tag{27}$$

Then applying Lemma 19 and Lemma 21, we obtain

$$\mathbb{E}\left\|\theta_{t+1}-\theta^*\right\|_2^2\le\frac{4L_{f_1,\theta}(K_{f_1}+K_{g_1})}{\lambda_\theta^2}\frac{\tau_\alpha^2}{1+t}+\frac{16R_\theta K_{f_1}}{\lambda_\theta^2(1+t)}+\frac{2L_{f_1,\theta}(K_{f_1}+K_{g_1})}{|\lambda_\theta|}\frac{\tau_\alpha\log(1+t)}{1+t}$$

$$+\frac{2(K_{f_1}^2+K_{g_1}^2)}{\lambda_\theta^2}\frac{1+\log(1+t)}{1+t}+\mathcal{O}(\frac{\log t}{t^\nu}+h(1,\nu))^{1-\epsilon'}. \tag{28}$$

### A.3 Technical Lemmas for Convergence Proof of Fast Time-scale Iteration

**Lemma 1.** *Let $p<0$, $0<q<1$, then for every integer $t\ge0$,*

$$\sum_{i=0}^t e^{p\sum_{k=i+1}^t(1+k)^{-q}}\frac{1}{(1+i)^{2q}}\le\frac{2e^{|p|/2}}{|p|}\left[D_p e^{p/2\sum_{k=0}^t(1+k)^{-q}}+\frac{1}{(1+t)^q}\right],$$

*where $D_p=e^{|p|/2\sum_{k=0}^{i_p}(1+k)^{-q}}$, with $i_p$ denoting a constant larger than $(|p|/2q)^{1/(1-q)}$.*

*Proof.* For detailed proof of Lemma 1 please refer to Theorem 4.3 in [7]. □

In order to bound the accumulated bias terms (14) and (15), we prove the following lemmas.

**Lemma 2.** *For any $\theta \in \mathbb{R}^d$ such that $\|\theta\|_2 \leq R_\theta$, $\|f_2(\theta, O_i)\|_2 \leq K_{f_2}$ for any $i \geq 0$, where $K_{f_2} < \infty$ is a bounded positive constant indepedent of $\theta$ and $w$.*

*Proof.* By the definition of $f_2(\theta, O_i)$, and denoting $\lambda_{cm} = \min|\lambda(C)|$, we obtain

$$
\begin{aligned}
||f_2(\theta, O_i)|| &= \left\|(A_i - C_iC^{-1}A)\theta + (b_i - C_iC^{-1}b)\right\|_2 \\
&\leq \left\|(A_i - C_iC^{-1}A)\theta\right\|_2 + \left\|(b_i - C_iC^{-1}b)\right\|_2 \\
&\leq \left(\|A_i\|_2 + \|C_i\|_2 \left\|C^{-1}\right\|_2 \|A\|_2\right)\|\theta\|_2 + \|b_i\|_2 + \|C_i\|_2 \left\|C^{-1}\right\|_2 \|b\|_2 \\
&\leq \left[(1+\gamma)\rho_{\max} + \frac{1}{\lambda_{cm}}(1+\gamma)\rho_{\max}\right]R_\theta + \rho_{\max}r_{\max} + \frac{1}{\lambda_{cm}}\rho_{\max}r_{\max} \\
&\triangleq K_{f_2}.
\end{aligned}
$$

$\square$

**Lemma 3.** *For all $\theta \in \mathbb{R}^d$ such that $\|\theta\|_2 \leq R_\theta$ and all $z \in \mathbb{R}^d$ such that $\|z\|_2 \leq R_w$, for all $i \geq 0$, (a) $\|\zeta_{f_2}(\theta, z, O_i)\|_2 \leq 4R_wK_{f_2}$; (b) $|\zeta_{f_2}(\theta_1, z_1, O_i) - \zeta_{f_2}(\theta_2, z_2, O_i)| \leq L_{f_2,\theta}\|\theta_1 - \theta_2\|_2 + L_{f_2,z}\|z_1 - z_2\|_2$.*

*Proof.* For (a), by the defination we have $\|\zeta_{f_2}(\theta, z, O_i)\|_2 = \|\langle f_2(\theta_t, O_t), z_t\rangle\|_2 \leq \|f_2(\theta, O_i)\|_2 \|z\|_2 \leq 2R_wK_{f_2}$.

For (b), we derive the bound as follows

$$
\begin{aligned}
|\zeta_{f_2}(\theta_1, z_1, O_i) - \zeta_{f_2}(\theta_2, z_2, O_i)| &= |\langle f_2(\theta_1, O_i), z_1\rangle - \langle f_2(\theta_2, O_i), z_2\rangle| \\
&\leq \|z_1\|_2 \|f_2(\theta_1, O_i) - f_2(\theta_2, O_i)\|_2 + \|f_2(\theta_2, O_i)\|_2 \|z_1 - z_2\|_2 \\
&\leq 2R_w \left\|(A_t - C_tC^{-1}A)(\theta_1 - \theta_2)\right\|_2 + 2K_{f_2}\|z_1 - z_2\|_2 \\
&\leq 2R_w\left[(1+\gamma)\rho_{\max} + \frac{1}{\lambda_{cm}}(1+\gamma)\rho_{\max}\right]\|\theta_1 - \theta_2\|_2 + 2K_{f_2}\|z_1 - z_2\|_2 \\
&\leq L_{f_2,\theta}\|\theta_1 - \theta_2\|_2 + L_{f_2,z}\|z_1 - z_2\|_2.
\end{aligned}
$$

$\square$

**Lemma 4.** *Let $K_{r_3} = [\max\{1, c_\alpha/c_\beta\}L_{f_2,\theta}(K_{f_1} + K_{g_1}) + L_{f_2,z}K_{r_2}]$. Then for $i \leq \tau_\beta$, $\mathbb{E}[\zeta_{f_2}(\theta_i, z_i, O_i)] \leq c_\beta K_{r_3}\tau_\beta$; and for $i > \tau_\beta$, $\mathbb{E}[\zeta_{f_2}(\theta_i, z_i, O_i)] \leq 8R_wK_{f_2}\beta_i + K_{r_3}\tau_\beta\beta_{i-\tau_\beta}$.*

*Proof.* Note that for any $i \geq 0$,

$$
\begin{aligned}
\|\theta_{i+1} - \theta_i\|_2 &= \|\Pi_{R_\theta}(\theta_i + \alpha_i(f_1(\theta_i, O_i) + g_1(z_i, O_i))) - \Pi_{R_\theta}\theta_i\|_2 \\
&\leq \|\theta_i + \alpha_i(f_1(\theta_i, O_i) + g_1(z_i, O_i)) - \theta_i\|_2 \\
&\leq \alpha_i \|f_1(\theta_i, O_i) + g_1(z_i, O_i)\|_2 \\
&\leq \alpha_i(K_{f_1} + K_{g_1}).
\end{aligned}
$$

Furthermore,

$$
\begin{aligned}
&\|z_{i+1} - z_i\|_2 \\
&= \left\|\Pi_{R_w}(z_i + \beta_i(f_2(\theta_i, O_i) + g_2(z_i, O_i)) - C^{-1}(b + A\theta_i)) + C^{-1}(b + A\theta_{i+1}) - z_i\right\|_2 \\
&= \left\|\Pi_{R_w}(z_i + \beta_i(f_2(\theta_i, O_i) + g_2(z_i, O_i)) - C^{-1}(b + A\theta_i)) + C^{-1}(b + A\theta_i) - z_i + C^{-1}A(\theta_{i+1} - \theta_i)\right\|_2 \\
&= \left\|\Pi_{R_w}(z_i + \beta_i(f_2(\theta_i, O_i) + g_2(z_i, O_i)) - C^{-1}(b + A\theta_i)) - \Pi_{R_w}[z_i - C^{-1}(b + A\theta_i)] + C^{-1}A(\theta_{i+1} - \theta_i)\right\|_2 \\
&\leq \left\|\Pi_{R_w}(z_i + \beta_i(f_2(\theta_i, O_i) + g_2(z_i, O_i)) - C^{-1}(b + A\theta_i)) - \Pi_{R_w}[z_i - C^{-1}(b + A\theta_i)]\right\|_2 \\
&\quad + \left\|C^{-1}A(\theta_{i+1} - \theta_i)\right\|_2 \\
&\leq \left\|(z_i + \beta_i(f_2(\theta_i, O_i) + g_2(z_i, O_i)) - C^{-1}(b + A\theta_i)) - [z_i - C^{-1}(b + A\theta_i)]\right\|_2 + \left\|C^{-1}A(\theta_{i+1} - \theta_i)\right\|_2 \\
&= \beta_i \|f_2(\theta_i, O_i) + g_2(z_i, O_i)\|_2 + \left\|C^{-1}A(\theta_{i+1} - \theta_i)\right\|_2 \\
&\leq \beta_i(K_{f_2} + K_{g_2}) + \alpha_i \left\|C^{-1}\right\|_2 \|A\|_2 (K_{f_1} + K_{g_1}) \\
&\leq \beta_i(K_{f_2} + K_{g_2} + \max\{1, \frac{c_\alpha}{c_\beta}\}\frac{(1+\gamma)\rho_{\max}}{\lambda_{cm}}(K_{f_1} + K_{g_1})) \\
&= \beta_i K_{r_2} \tag{29}
\end{aligned}
$$

where $K_{r_2} \triangleq K_{f_2} + K_{g_2} + \max\{1, \frac{c_\alpha}{c_\beta}\}\frac{(1+\gamma)\rho_{\max}}{\lambda_{cm}}(K_{f_1} + K_{g_1})$. Applying the Lipschitz continuous property in Lemma 3, it follows that

$$|\zeta_{f_2}(\theta_i, z_i, O_i) - \zeta_{f_2}(\theta_{i-\tau}, z_{i-\tau}, O_i)| \leq L_{f_2,\theta}\|\theta_i - \theta_{i-\tau}\|_2 + L_{f_2,z}\|z_i - z_{i-\tau}\|_2$$

$$\leq L_{f_2,\theta}(K_{f_1} + K_{g_1})\sum_{k=i-\tau}^{i-1}\alpha_k + L_{f_2,z}K_{r_2}\sum_{k=i-\tau}^{i-1}\beta_k.$$

The next step is to provide an upper bound for $\mathbb{E}[\zeta_{f_2}(\theta_{i-\tau}, z_{i-\tau}, O_i)]$. We further define an independent $(\theta'_{i-\tau}, z'_{i-\tau})$ and $O'_i = (s'_i, a'_i, r'_i, s'_{i+1})$ that has the same marginal distribution as $(\theta_{i-\tau}, z_{i-\tau})$ and $O_i$. It is clear that $\mathbb{E}[\zeta_{f_2}(\theta'_{i-\tau}, z'_{i-\tau}, O'_i)] = 0$. Note that the following Markov chain holds

$$(\theta_{i-\tau}, z_{i-\tau}) \to s_{i-\tau} \to s_i \to O_i.$$

Since $\|\zeta_{f_2}(\theta, z, O_i)\|_2 \leq 4R_w K_{f_2}$ for all $\theta, z \in \mathbb{R}^d$, by Lemma 3, applying Lemma 10 in [3] yields

$$\mathbb{E}[\zeta_{f_2}(\theta_{i-\tau}, z_{i-\tau}, O_i)] \leq |\mathbb{E}[\zeta_{f_2}(\theta_{i-\tau}, z_{i-\tau}, O_i)] - \mathbb{E}[\zeta_{f_2}(\theta'_{i-\tau}, z'_{i-\tau}, O'_i)]| \leq 8R_w K_{f_2}m\rho^\tau.$$

Recall that $\tau_\beta = \min\{i \geq 0 : m\rho^i \leq \beta_t\}$. For $i \leq \tau_\beta$, it follows that

$$\mathbb{E}[\zeta_{f_2}(\theta_i, z_i, O_i)] \leq \mathbb{E}[\zeta_{f_2}(\theta_0, z_0, O_i)] + L_{f_2,\theta}(K_{f_1} + K_{g_1})\sum_{k=0}^{i-1}\alpha_k + L_{f_2,z}K_{r_2}\sum_{k=0}^{i-1}\beta_k$$

$$\leq L_{f_2,\theta}(K_{f_1} + K_{g_1})i\alpha_0 + L_{f_2,z}K_{r_2}i\beta_0$$

$$\leq c_\beta[\max\{1, \frac{c_\alpha}{c_\beta}\}L_{f_2,\theta}(K_{f_1} + K_{g_1}) + L_{f_2,z}K_{r_2}]\tau_\beta$$

$$\leq c_\beta K_{r_3}\tau_\beta.$$

For $i > \tau_\beta$, it follows that

$$\mathbb{E}[\zeta_{f_2}(\theta_i, z_i, O_i)] \leq \mathbb{E}[\zeta_{f_2}(\theta_{i-\tau_\beta}, z_{i-\tau_\beta}, O_i)] + L_{f_2,\theta}(K_{f_1} + K_{g_1})\sum_{k=i-\tau_\beta}^{i-1}\alpha_k + L_{f_2,z}K_{r_2}\sum_{k=i-\tau_\beta}^{i-1}\beta_k$$

$$\leq 8R_w K_{f_2}m\rho^{\tau_\beta} + L_{f_2,\theta}(K_{f_1} + K_{g_1})\tau_\beta\alpha_{i-\tau_\beta} + L_{f_2,z}K_{r_2}\tau_\beta\beta_{i-\tau_\beta}$$

$$\leq 8R_w K_{f_2}\beta_t + [\max\{1, \frac{c_\alpha}{c_\beta}\}L_{f_2,\theta}(K_{f_1} + K_{g_1}) + L_{f_2,z}K_{r_2}]\tau_\beta\beta_{i-\tau_\beta}$$

$$= 8R_w K_{f_2}\beta_t + K_{r_3}\tau_\beta\beta_{i-\tau_\beta}.$$

$\square$

**Lemma 5.** *Fix $0 < \nu < 1$, and let $\beta_t = c_\beta/(1+t)^\nu$. Then*

$$\sum_{i=0}^{t} e^{\lambda_w \sum_{k=i+1}^{t}\beta_k}\beta_i\mathbb{E}[\zeta_{f_2}(\theta_i, z_i, O_i)]$$

$$\leq c_\beta K_{r_3}\tau_\beta\frac{e^{-\lambda_w c_\beta}}{-\lambda_w}e^{\frac{\lambda_w c_\beta}{1-\nu}[(1+t)^\nu - (1+\tau_\beta)^\nu]} + 8R_w K_{f_2}\frac{e^{-\lambda_w c_\beta}}{-\lambda_w}\frac{c_\beta}{(1+t)^\nu}$$

$$+ 2K_{r_3}\tau_\beta\frac{e^{-\lambda_w c_\beta/2}}{-\lambda_w}(e^{\frac{-\lambda_w c_\beta}{2(1-\nu)}[(t+1)^{1-\nu}-1]}D_1\mathbb{1}_{\{\tau_\beta+1<i_{d_1}\}} + \beta_{t-\tau_\beta}),$$

*where $D_1 = c_\beta \max_{i\in[0,i_{d_1}]}\{e^{-(\lambda_w/2)\sum_{k=0}^{i}\beta_k}\}$ and $i_{d_1} = (\frac{-2\nu}{\lambda_w c_\beta})^{\frac{1}{1-\nu}}$.*

*Proof.* Applying Lemma 4, it follows that

$$\sum_{i=0}^{t} e^{\lambda_w \sum_{k=i+1}^{t}\beta_k}\beta_i\mathbb{E}[\zeta_{f_2}(\theta_i, z_i, O_i)]$$

$$\leq c_\beta K_{r_3}\tau_\beta\sum_{i=0}^{\tau_\beta} e^{\lambda_w \sum_{k=i+1}^{t}\beta_k}\beta_i + 8R_w K_{f_2}\beta_t\sum_{i=\tau_\beta+1}^{t} e^{\lambda_w \sum_{k=i+1}^{t}\beta_k}\beta_i$$

$$+ K_{r_3}\tau_\beta\sum_{i=\tau_\beta+1}^{t} e^{\lambda_w \sum_{k=i+1}^{t}\beta_k}\beta_{i-\tau_\beta}\beta_i. \tag{30}$$

For the first term in (30), we have

$$\sum_{i=0}^{\tau_\beta} e^{\lambda_w \sum_{k=i+1}^{t} \beta_k} \beta_i \leq \max_{i \geq 0}\{e^{-\lambda_w \beta_i}\} \sum_{i=0}^{\tau_\beta} e^{\lambda_w \sum_{k=i}^{t} \beta_k} \beta_i$$

$$= e^{-\lambda_w c_\beta} \sum_{i=0}^{\tau_\beta} e^{\lambda_w (T_{t+1}-T_i)} \beta_i$$

$$\leq e^{-\lambda_w c_\beta} \int_0^{T_{\tau_\beta+1}} e^{\lambda_w (T_{t+1}-s)} ds$$

$$\leq \frac{e^{-\lambda_w c_\beta}}{-\lambda_w} e^{\lambda_w (T_{t+1}-T_{\tau_\beta+1})}$$

$$\leq \frac{e^{-\lambda_w c_\beta}}{-\lambda_w} e^{\lambda_w c_\beta \sum_{k=\tau_\beta}^{t} 1/(1+k)^{-\nu}}$$

$$= \frac{e^{-\lambda_w c_\beta}}{-\lambda_w} e^{\frac{\lambda_w c_\beta}{1-\nu}[(1+t)^\nu - (1+\tau_\beta)^\nu]}, \tag{31}$$

where $T_n = \sum_{k=0}^{n-1} \beta_k$. For the second term in (30), we have

$$\beta_t \sum_{i=\tau_\beta+1}^{t} e^{\lambda_w \sum_{k=i+1}^{t} \beta_k} \beta_i \leq \max_{i \geq 0}\{e^{-\lambda_w \beta_i}\} \beta_t \sum_{i=\tau_\beta+1}^{t} e^{\lambda_w \sum_{k=i+1}^{t} \beta_k} \beta_i$$

$$\leq e^{-\lambda_w c_\beta} \beta_t \sum_{i=\tau_\beta+1}^{t} e^{\lambda_w (T_{t+1}-T_i)} \beta_i$$

$$\leq e^{-\lambda_w c_\beta} \beta_t \int_{T_{\tau_\beta+1}}^{T_{t+1}} e^{\lambda_w (T_{t+1}-s)} ds$$

$$= \frac{e^{-\lambda_w c_\beta}}{-\lambda_w} \beta_t \left(1 - e^{\lambda_w (T_{t+1}-T_{\tau_\beta+1})}\right)$$

$$\leq \frac{e^{-\lambda_w c_\beta}}{-\lambda_w} \frac{c_\beta}{(1+t)^\nu}. \tag{32}$$

For the third term in (30), we have

$$\sum_{i=\tau_\beta+1}^{t} e^{\lambda_w \sum_{k=i+1}^{t} \beta_k} \beta_{i-\tau_\beta} \beta_i \leq \max_{i \in [\tau_\beta+1,t]}\{e^{(\lambda_w/2) \sum_{k=i+1}^{t} \beta_k} \beta_{i-\tau_\beta}\} \sum_{i=\tau_\beta+1}^{t} e^{(\lambda_w/2) \sum_{k=i+1}^{t} \beta_k} \beta_i$$

$$\leq \max_{i \in [\tau_\beta+1,t]}\{e^{(\lambda_w/2) \sum_{k=i+1}^{t} \beta_k} \beta_{i-\tau_\beta}\} \frac{2 e^{-\lambda_w c_\beta/2}}{-\lambda_w}. \tag{33}$$

To bound (33), we define $y_i = e^{(\lambda_w/2) \sum_{k=i+1}^{t} \beta_k} \beta_{i-\tau_\beta}$, and then we have

$$\frac{y_{i+1}}{y_i} = e^{-(\lambda_w/2)\beta_{i+1}} \left(1 - \frac{1}{2+i-\tau_\beta}\right)^\nu.$$

If $i \geq i_{d_1}$ and $\tau_\beta + 1 > i_{d_1}$, then $\frac{y_{i+1}}{y_i} \geq 1$ for all $i \in [\tau_\beta + 1, t]$. Thus

$$\max_{i \in [\tau_\beta+1,t]}\{e^{(\lambda_w/2) \sum_{k=i+1}^{t} \beta_k} \beta_{i-\tau_\beta}\} = \beta_{t-\tau_\beta}. \tag{34}$$

If $\tau_\beta + 1 < i_{d_1}$, then

$$\max_{i \in [\tau_\beta + 1, t]} \{e^{(\lambda_w/2) \sum_{k=i+1}^t \beta_k} \beta_{i-\tau_\beta}\}$$

$$\leq \max_{i \in [\tau_\beta + 1, i_{d_1}]} \{e^{(\lambda_w/2) \sum_{k=i+1}^t \beta_k} \beta_{i-\tau_\beta}\} + \max_{i \in [i_{d_1}+1, t]} \{e^{(\lambda_w/2) \sum_{k=i+1}^t \beta_k} \beta_{i-\tau_\beta}\}$$

$$\leq e^{(\lambda_w/2) \sum_{k=0}^t \beta_k} \max_{i \in [\tau_\beta+1, i_{d_1}]} \{e^{-(\lambda_w/2) \sum_{k=0}^i \beta_k} \beta_{i-\tau_\beta}\} + \beta_{t-\tau_\beta}$$

$$\leq e^{(\lambda_w/2) \sum_{k=0}^t \beta_k} \max_{i \in [0, i_{d_1}]} \{e^{-(\lambda_w/2) \sum_{k=0}^i \beta_k} \beta_0\} + \beta_{t-\tau_\beta}$$

$$\leq e^{\frac{\lambda_w c_\beta}{2(1-\nu)}[(t+1)^{1-\nu}-1]} D_1 + \beta_{t-\tau_\beta}. \tag{35}$$

Combining (34) and (35) and substituting into (33), we have

$$\sum_{i=\tau_\beta+1}^t e^{\lambda_w \sum_{k=i+1}^t \beta_k} \beta_{i-\tau_\beta} \beta_i \leq \frac{2e^{-\lambda_w c_\beta/2}}{-\lambda_w} (e^{\frac{\lambda_w c_\beta}{2(1-\nu)}[(t+1)^{1-\nu}-1]} D_1 \mathbb{1}_{\{\tau_\beta+1<i_\beta\}} + \beta_{t-\tau_\beta}). \tag{36}$$

Finally, (36), (32), and (35) imply that

$$\sum_{i=0}^t e^{\lambda_w \sum_{k=i+1}^t \beta_k} \beta_i \mathbb{E}[\zeta_{f_2}(\theta_i, z_i, O_i)]$$

$$\leq [c_\alpha L_{f_2, \theta}(K_{f_1} + K_{g_1}) + c_\beta L_{f_2, z} K_{r_2}] \tau_\beta \frac{e^{-\lambda_w c_\beta}}{-\lambda_w} e^{\frac{\lambda_w c_\beta}{1-\nu}[(1+t)^\nu - (1+\tau_\beta)^\nu]} \tag{37}$$

$$+ 4R_w K_{f_2} \frac{e^{-\lambda_w c_\beta}}{-\lambda_w} \frac{c_\beta}{(1+t)^\nu} + 2K_{r_3} \tau_\beta \frac{e^{-\lambda_w c_\beta/2}}{-\lambda_w} (e^{\frac{-\lambda_w c_\beta}{2(1-\nu)}[(t+1)^{1-\nu}-1]} D_1 \mathbb{1}_{\{\tau_\beta+1<i_{d_1}\}} + \beta_{t-\tau_\beta}).$$
$$\tag{38}$$

$\square$

**Lemma 6.** *For any $z \in \mathbb{R}^d$ such that $\|z\|_2 \leq R_w$, $\|g_2(z, O_i)\|_2 \leq K_{g_2}$ for any $i \geq 0$.*

*Proof.* By the definition of $g_2(z, O_t)$, we obtain

$$\|g_2(z, O_i)\|_2 = \|C_i z_i\|_2 \leq \|C_i\|_2 \|z_i\|_2 \leq 2R_w \leq K_{g_2}.$$

$\square$

**Lemma 7.** *For all $z \in \mathbb{R}^d$ such that $\|z\|_2 \leq R_w$, we have for all $i \geq 0$, (1) $\|\zeta_{g_2}(z, O_i)\|_2 \leq 4R_w K_{g_2}$; (2) $|\zeta_{g_2}(z_1, O_i) - \zeta_{g_2}(z_2, O_i)| \leq L_{g_2, z} \|z_1 - z_2\|_2$.*

*Proof.* For (1), by the defination of $\zeta_{g_2}(z, O_i)$, we have $\|\zeta_{g_2}(z_i, O_i)\|_2 = \|\langle g_2(z_t, O_t) - \bar{g}_2(z_t), z_t \rangle\|_2 \leq (\|g_2(\theta_i, O_i)\|_2 + \|\bar{g}_2(\theta_i)\|_2) \|z_i\|_2 \leq 4R_w K_{g_2}$. For (2), we derive the bound as follows.

$$|\zeta_{g_2}(z_1, O_i) - \zeta_{g_2}(z_2, O_i)|$$
$$= |\langle g_2(z_1, O_i) - \bar{g}_2(z_1), z_1 \rangle + \langle g_2(z_2, O_i) - \bar{g}_2(z_2), z_2 \rangle|$$
$$\leq \|z_1\|_2 \|g_2(z_1, O_i) - \bar{g}_2(z_1) - g_2(z_2, O_i) + \bar{g}_2(z_2)\|_2 + \|g_2(z_2, O_i) - \bar{g}_2(z_2)\|_2 \|z_1 - z_2\|_2$$
$$= \|z_1\|_2 \|(C_i - C)(z_1 - z_2)\|_2 + \|g_2(z_2, O_i) - \bar{g}_2(z_2)\|_2 \|z_1 - z_2\|_2$$
$$\leq 2R_w(\|C_i\|_2 + \|C\|_2) \|z_1 - z_2\|_2 + 2K_{g_2} \|z_1 - z_2\|_2$$
$$\leq 4R_w \|z_1 - z_2\|_2 + 2K_{g_2} \|z_1 - z_2\|_2$$
$$\leq L_{g_2, z} \|z_1 - z_2\|_2.$$

$\square$

**Lemma 8.** *For $i \leq \tau_\beta$, $\mathbb{E}[\zeta_{g_2}(z_i, O_i)] \leq c_\beta L_{g_2, z} K_{r_2} \tau_\beta$; and for $i > \tau_\beta$, $\mathbb{E}[\zeta_{g_2}(z_i, O_i)] \leq 8R_w K_{g_2} \beta_t + L_{g_2, z} K_{r_2} \tau_\beta \beta_{i-\tau_\beta}$.*

*Proof.* Applying the Lipschitz continuous property of $\zeta_{g_2}(z, O_i)$ and the inequality (29) in Lemma 4, it follows that

$$|\zeta_{g_2}(z_i, O_i) - \zeta_{g_2}(z_{i-\tau}, O_i)| \le L_{g_2,z} \|z_i - z_{i-\tau}\|_2 \le L_{g_2,z} K_{r_2} \sum_{k=i-\tau}^{i-1} \beta_k.$$

Then we need to provide an upper bound for $\mathbb{E}[\zeta_{g_2}(z_{i-\tau}, O_i)]$. We further define an independent $z'_{i-\tau}$ and $O'_i = (s'_i, a'_i, r'_i, s'_{i+1})$ which have the same marginal distribution as $z_{i-\tau}$ and $O_i$. Using Lemma 7 and following the steps similar to those in Lemma 4, we obtain

$$\mathbb{E}[\zeta_{g_2}(z_{i-\tau}, O_i)] \le |\mathbb{E}[\zeta_{g_2}(z_{i-\tau}, O_i)] - \mathbb{E}[\zeta_{f_2}(z'_{i-\tau}, O'_i)]| \le 8 R_w K_{g_2} m \rho^\tau.$$

For $i \le \tau_\beta$, it follows that

$$\mathbb{E}[\zeta_{g_2}(z_i, O_i)] \le \mathbb{E}[\zeta_{g_2}(z_0, O_i)] + L_{g_2,z} K_{r_2} \sum_{k=0}^{i-1} \beta_k \le L_{g_2,z} K_{r_2} i \beta_0 \le c_\beta L_{g_2,z} K_{r_2} \tau_\beta.$$

For $i > \tau_\beta$, it follows that

$$\mathbb{E}[\zeta_{g_2}(z_i, O_i)] \le \mathbb{E}[\zeta_{g_2}(z_{i-\tau_\beta}, O_i)] + L_{g_2,z} K_{r_2} \sum_{k=i-\tau_\beta}^{i-1} \beta_k$$

$$\le 8 R_w K_{g_2} m \rho^{\tau_\beta} + L_{g_2,z} K_{r_2} \tau_\beta \beta_{i-\tau_\beta}$$

$$\le 8 R_w K_{g_2} \beta_t + L_{g_2,z} K_{r_2} \tau_\beta \beta_{i-\tau_\beta}.$$

$\square$

**Lemma 9.** *Fix $0 < \nu < 1$, and let $\beta_t = c_\beta/(1+t)^\nu$. Then*

$$\sum_{i=0}^{t} e^{\lambda_w \sum_{k=i+1}^{t} \beta_k} \beta_i \mathbb{E}[\zeta_{g_2}(z_i, O_i)]$$

$$\le c_\beta L_{g_2,z} K_{r_2} \tau_\beta \frac{e^{-\lambda_w c_\beta}}{-\lambda_w} e^{\frac{\lambda_w c_\beta}{1-\nu}[(1+t)^\nu - (1+\tau_\beta)^\nu]} + 8 R_w K_{g_2} \frac{e^{-\lambda_w c_\beta}}{-\lambda_w} \frac{c_\beta}{(1+t)^\nu}$$

$$+ 2 L_{g_2,z} K_{r_2} \tau_\beta \frac{e^{-\lambda_w c_\beta/2}}{-\lambda_w} \left( e^{\frac{\lambda_w c_\beta}{2(1-\nu)}[(t+1)^{1-\nu}-1]} D_1 \mathbb{1}_{\{\tau_\beta+1 < i_{d_1}\}} + \beta_{t-\tau_\beta} \right).$$

*where $D_1 = c_\beta \max_{i \in [0, i_{d_1}]}\{ e^{-(\lambda_w/2) \sum_{k=0}^{i} \beta_k} \}$ and $i_{d_1} = (\frac{-2\nu}{\lambda_w c_\beta})^{\frac{1}{1-\nu}}$.*

*Proof.* Applying Lemma 8, it follows that

$$\sum_{i=0}^{t} e^{\lambda_w \sum_{k=i+1}^{t} \beta_k} \beta_i \mathbb{E}[\zeta_{g_2}(z_i, O_i)]$$

$$\le c_\beta L_{g_2,z} K_{r_2} \tau_\beta \sum_{i=0}^{\tau_\beta} e^{\lambda_w \sum_{k=i+1}^{t} \beta_k} \beta_i + 8 R_w K_{g_2} \beta_t \sum_{i=\tau_\beta+1}^{t} e^{\lambda_w \sum_{k=i+1}^{t} \beta_k} \beta_i$$

$$+ L_{g_2,z} K_{r_2} \tau_\beta \sum_{i=\tau_\beta+1}^{t} e^{\lambda_w \sum_{k=i+1}^{t} \beta_k} \beta_{i-\tau_\beta} \beta_i.$$

Following steps similar to those in (30)-(37), we have the desired result. $\square$

**Lemma 10.** *For given $0 < \nu < \sigma < 1$, let $\beta_t = c_\beta/(1+t)^\nu$ and $\alpha_t = c_\alpha/(1+t)^\sigma$. Then*

$$\sum_{i=0}^{t} e^{\lambda_w \sum_{k=i+1}^{t} \beta_k} \mathbb{E}\langle C^{-1} A(\theta_{i+1} - \theta_i), z_i \rangle$$

$$\le \frac{2 c_\alpha (1+\gamma) \rho_{\max}}{c_\beta \lambda_{cm}} R_w (K_{f_1} + K_{g_1}) \frac{2 e^{-\lambda_w c_\beta/2}}{-\lambda_w} \left( e^{\frac{\lambda_w c_\beta}{2(1-\nu)}[(1+t)^{1-\nu}-1]} D_2 + \frac{1}{(1+t)^{\sigma-\nu}} \right),$$

*where $D_2 = \max_{i \in [0, i_{d_2}]}\{ e^{-(\lambda_w/2) \sum_{k=0}^{i} \beta_k} \frac{1}{(1+i)^{\sigma-\nu}} \}$ and $i_{d_2} = (\frac{-2(\sigma-\nu)}{\lambda_w c_\beta})^{\frac{1}{1-\nu}}$.*

*Proof.* Applying Lemmas 13 and 12, it follows that

$$\sum_{i=0}^{t} e^{\lambda_w \sum_{k=i+1}^{t} \beta_k} \mathbb{E}\langle C^{-1} A(\theta_{i+1} - \theta_i), z_i \rangle$$

$$\leq \sum_{i=0}^{t} e^{\lambda_w \sum_{k=i+1}^{t} \beta_k} \mathbb{E}\left[ \left\| C^{-1} \right\|_2 \|A\|_2 \|\theta_{i+1} - \theta_i\|_2 \|z_i\|_2 \right]$$

$$\leq 2 \left\| C^{-1} \right\|_2 \|A\|_2 R_w (K_{f_1} + K_{g_1}) \sum_{i=0}^{t} e^{\lambda_w \sum_{k=i+1}^{t} \beta_k} \alpha_i$$

$$\leq \frac{2(1+\gamma)\rho_{\max}}{\lambda_{cm}} R_w (K_{f_1} + K_{g_1}) \sum_{i=0}^{t} e^{\lambda_w \sum_{k=i+1}^{t} \beta_k} \beta_i \frac{\alpha_i}{\beta_i}$$

$$\leq \frac{2c_\alpha(1+\gamma)\rho_{\max}}{c_\beta \lambda_{cm}} R_w (K_{f_1} + K_{g_1}) \max_{i \in [0,t]} \{ e^{(\lambda_w/2) \sum_{k=i+1}^{t} \beta_k} \frac{1}{(1+i)^{\sigma-\nu}} \} \sum_{i=0}^{t} e^{(\lambda_w/2) \sum_{k=i+1}^{t} \beta_k} \beta_i$$

$$\leq \frac{2c_\alpha(1+\gamma)\rho_{\max}}{c_\beta \lambda_{cm}} R_w (K_{f_1} + K_{g_1}) \frac{2e^{-\lambda_w c_\beta/2}}{-\lambda_w} \left( e^{\frac{\lambda_w c_\beta}{2(1-\nu)}[(1+t)^{1-\nu}-1]} D_2 + \frac{1}{(1+t)^{\sigma-\nu}} \right). \qquad (39)$$

Based on (39), we follow similar steps in Theroem 4.3 [7] and obtain the following upper bound

$$\sum_{i=0}^{t} e^{(\lambda_w/2) \sum_{k=i+1}^{t} \beta_k} \beta_i \leq \frac{2e^{-\lambda_w C_\beta/2}}{-\lambda_w}, \qquad (40)$$

and

$$\max_{i \in [0,t]} \{ e^{(\lambda_w/2) \sum_{k=i+1}^{t} \beta_k} \frac{1}{(1+i)^{\sigma-\nu}} \} \leq e^{\frac{\lambda_w c_\beta}{2(1-\nu)}[(1+t)^{1-\nu}-1]} D_2 + \frac{1}{(1+t)^{\sigma-\nu}}, \qquad (41)$$

where $D_2 = \max_{i \in [0, i_{d_2}]} \{ e^{-(\lambda_w/2) \sum_{k=0}^{i} \beta_k} \frac{1}{(1+i)^{\sigma-\nu}} \}$ and $i_{d_2} = (\frac{-2(\sigma-\nu)}{\lambda_w c_\beta})^{\frac{1}{1-\nu}}$. □

**Lemma 11.** *Suppose* (18) *holds. If* $\sigma > \frac{3}{2}\nu$, *we have*

$$\sum_{i=0}^{t} e^{\lambda_w \sum_{k=i+1}^{t} \beta_k} \mathbb{E}\langle C^{-1} A(\theta_{i+1} - \theta_i), z_i \rangle = \mathcal{O}\left( \frac{1}{t^\nu} \right),$$

*amd if* $\nu < \sigma \leq \frac{3}{2}\nu$, *we have*

$$\sum_{i=0}^{t} e^{\lambda_w \sum_{k=i+1}^{t} \beta_k} \mathbb{E}\langle C^{-1} A(\theta_{i+1} - \theta_i), z_i \rangle = \mathcal{O}\left( \frac{1}{t^{2(\sigma-\nu)-\epsilon}} \right),$$

*where* $\epsilon$ *is any constant in* $(0, \sigma - \nu]$.

*Proof.* If $\sigma \geq 2\nu$, (10) implies that

$$\sum_{i=0}^{t} e^{\lambda_w \sum_{k=i+1}^{t} \beta_k} \mathbb{E}\langle C^{-1} A(\theta_{i+1} - \theta_i), z_i \rangle = \mathcal{O}\left( \frac{1}{(1+t)^\nu} \right).$$

If $\sigma \leq 2\nu$, it follows that $\mathbb{E}\|z_t\|_2^2 = \mathcal{O}(\frac{1}{t^{\sigma-\nu}})$. Hence there exists a constant $0 < C < \infty$ and $T > 0$ such that

$$\mathbb{E}\|z_t\|_2^2 \leq 4R_w^2 \qquad \text{for all } 0 \leq t \leq T, \qquad (42)$$

$$\mathbb{E}\|z_t\|_2^2 \leq \frac{C}{(1+t)^{(\sigma-\nu)}} \qquad \text{for all } t > T. \qquad (43)$$

Then, substituting (42) and (43) into (16), we have

$$\sum_{i=0}^{t} e^{\lambda_w \sum_{k=i+1}^{t} \beta_k} \mathbb{E}\langle C^{-1} A(\theta_{i+1} - \theta_i), z_i \rangle \tag{44}$$

$$\leq \left\| C^{-1} \right\|_2 \|A\|_2 (K_{f_1} + K_{g_1}) \sum_{i=0}^{t} e^{\lambda_w \sum_{k=i+1}^{t} \beta_k} \alpha_i \sqrt{\mathbb{E}\|z_i\|_2^2} \tag{45}$$

$$\leq \frac{(1+\gamma)\rho_{\max}}{\lambda_{cm}} (K_{f_1} + K_{g_1}) \Big( \sum_{i=0}^{T} e^{\lambda_w \sum_{k=i+1}^{t} \beta_k} \alpha_i \sqrt{\mathbb{E}\|z_i\|_2^2} + \sum_{i=T+1}^{t} e^{\lambda_w \sum_{k=i+1}^{t} \beta_k} \alpha_i \sqrt{\mathbb{E}\|z_i\|_2^2} \Big) \tag{46}$$

$$\leq \frac{c_\alpha (1+\gamma)\rho_{\max}}{c_\beta \lambda_{cm}} (K_{f_1} + K_{g_1}) \Big( 2R_w \sum_{i=0}^{T} e^{\lambda_w \sum_{k=i+1}^{t} \beta_k} \beta_i \frac{1}{(1+i)^{(\sigma-\nu)}}$$
$$+ C \sum_{i=T+1}^{t} e^{\lambda_w \sum_{k=i+1}^{t} \beta_k} \beta_i \frac{1}{(1+i)^{1.5(\sigma-\nu)}} \Big). \tag{47}$$

Here, we follow similar steps in (31) and (33)-(36) to get

$$\sum_{i=0}^{T} e^{\lambda_w \sum_{k=i+1}^{t} \beta_k} \beta_i \frac{1}{(1+i)^{(\sigma-\nu)}} \leq \sum_{i=0}^{T} e^{\lambda_w \sum_{k=i+1}^{t} \beta_k} \beta_i \leq \frac{e^{-\lambda_w c_\beta}}{-\lambda_w} e^{\frac{\lambda_w c_\beta}{1-\nu} [(1+t)^\nu - (1+T)^\nu]},$$

and

$$\sum_{i=T+1}^{t} e^{\lambda_w \sum_{k=i+1}^{t} \beta_k} \beta_i \frac{1}{(1+i)^{1.5(\sigma-\nu)}} \leq \frac{2e^{-\lambda_w C_\beta/2}}{-\lambda_w} \Big( e^{\frac{\lambda_w c_\beta}{2(1-\nu)} [(1+t)^{1-\nu} - 1]} D + \frac{1}{(1+t)^{1.5(\sigma-\nu)}} \Big),$$

where $D = \max_{i \in [0, i_d]} \{ e^{-(\lambda_w/2) \sum_{k=0}^{i} \beta_k} \frac{1}{(1+i)^{1.5(\sigma-\nu)}} \}$ and $i_d = (\frac{-3(\sigma-\nu)}{\lambda_w c_\beta})^{\frac{1}{1-\nu}}$.

It follows that

$$\sum_{i=0}^{t} e^{\lambda_w \sum_{k=i+1}^{t} \beta_k} \alpha_i \mathbb{E}\langle C^{-1} A(\theta_{i+1} - \theta_i), z_i \rangle = \mathcal{O}\left( \frac{1}{t^{1.5(\sigma-\nu)}} \right).$$

If $\frac{3}{2}\nu < \sigma \leq 2\nu$, we have $\mathbb{E}\|z_t\|_2^2 = \mathcal{O}\left( \frac{1}{t^{1.5(\sigma-\nu)}} \right)$. Then, by following the similar steps in (42)-(47), we have

$$\sum_{i=0}^{t} e^{\lambda_w \sum_{k=i+1}^{t} \beta_k} \alpha_i \mathbb{E}\langle C^{-1} A(\theta_{i+1} - \theta_i), z_i \rangle = \mathcal{O}\left( \frac{1}{t^{1.75(\sigma-\nu)}} \right),$$

and $\mathbb{E}\|z_t\|_2^2 = \mathcal{O}\left( \frac{1}{t^{1.75(\sigma-\nu)}} \right)$. Then we repeat the steps (42)-(47) for a total number $N = \lceil -\log_2(2 - \frac{\nu}{\sigma-\nu}) \rceil$ of times, we have

$$\sum_{i=0}^{t} e^{\lambda_w \sum_{k=i+1}^{t} \beta_k} \alpha_i \mathbb{E}\langle C^{-1} A(\theta_{i+1} - \theta_i), z_i \rangle = \mathcal{O}\left( \frac{1}{t^{(2-2^{-N})(\sigma-\nu)}} \right) = \mathcal{O}\left( \frac{1}{(1+t)^\nu} \right).$$

Since $(2 - 2^{-N})(\sigma - \nu) > \nu$, we have $\mathbb{E}\|z_t\|_2^2 = \mathcal{O}(\frac{\log t}{t^\nu}) + \mathcal{O}(\frac{1}{t^\nu})$.

If $\nu < \sigma \leq \frac{3}{2}\nu$, then we repeat steps (42)-(47) for a total number $N = \lceil \log_2(\frac{\sigma-\nu}{\epsilon}) \rceil$ of times, we have

$$\sum_{i=0}^{t} e^{\lambda_w \sum_{k=i+1}^{t} \beta_k} \alpha_i \mathbb{E}\langle C^{-1} A(\theta_{i+1} - \theta_i), z_i \rangle = \mathcal{O}\left( \frac{1}{(1+t)^{2(\sigma-\nu)-\epsilon}} \right).$$

$\square$

### A.4 Technical Lemmas for Convergence Proof of Slow Time-scale Iteration

In this subsection, we obtain the following properties for the slow time-scale.

**Lemma 12.** *For any $\theta \in \mathbb{R}^d$ such that $\|\theta\|_2 \leq R_\theta$, $\|f_1(\theta, O_i)\|_2 \leq K_{f_1}$ for any $i \geq 0$, where $K_{f_1} < \infty$ is a bounded constant indepedent of $\theta$ and $w$.*

*Proof.* By the definition of $f_1(\theta, O_i)$, and denoting $\lambda_{cm} = \min |\lambda(C)|$, we obtain

$$
\begin{aligned}
\|f_1(\theta, O_i)\| &= \left\|(A_i - B_i C^{-1} A)\theta + (b_i - B_i C^{-1} b)\right\|_2 \\
&\leq \left\|(A_i - B_i C^{-1} A)\theta\right\|_2 + \left\|(b_i - B_i C^{-1} b)\right\|_2 \\
&\leq (\|A_i\|_2 + \|B_i\|_2 \|C^{-1}\|_2 \|A\|_2)\|\theta\|_2 + \|b_i\|_2 + \|B_i\|_2 \|C^{-1}\|_2 \|b\|_2 \\
&\leq \left[(1 + \gamma)\rho_{\max} + \frac{1}{\lambda_{cm}}\gamma(1 + \gamma)\rho_{\max}^2\right] + \rho_{\max} r_{\max} + \frac{1}{\lambda_{cm}}\gamma\rho_{\max}^2 r_{\max} \\
&\leq K_{f_1}.
\end{aligned}
$$

$\square$

**Lemma 13.** *For any $z \in \mathbb{R}^d$ such that $\|z\|_2 \leq 2R_w$, $\|g_1(z, O_i)\|_2 \leq K_{g_1}$ for any $i \geq 0$.*

*Proof.* By the definition of $g_1(z, O_i)$, we obtain $\|g_1(z_t, O_t)\|_2 = \|B_t z_t\|_2 \leq \|B_t\|_2 \|z_t\|_2 \leq 2\gamma\rho_{\max} R_w$. $\square$

**Lemma 14.** *For all $\theta \in \mathbb{R}^d$ such that $\|\theta\|_2 \leq R_\theta$, we have for all $i \geq 0$, (a) $\|\zeta_{f_1}(\theta, O_i)\|_2 \leq 4R_\theta K_{f_1}$; (b) $|\zeta_{f_2}(\theta_1, O_i) - \zeta_{f_2}(\theta_2, O_i)| \leq L_{f_1, \theta}\|\theta_1 - \theta_2\|_2$.*

*Proof.* For (a), following steps similar in (12), we have $\left\|\bar{f}_1(\theta)\right\|_2 \leq K_{f_1}$. Then by the defination we have

$$
\|\zeta_{f_1}(\theta, O_i)\|_2 \leq (\|f_1(\theta, O_i)\|_2 + \left\|\bar{f}_1(\theta)\right\|_2)(\|\theta\|_2 + \|\theta^*\|_2) \leq 4R_\theta K_{f_1}.
$$

For (b), we derive the bound as follows

$$
\begin{aligned}
&|\zeta_{f_1}(\theta_1, O_i) - \zeta_{f_1}(\theta_2, O_i)| \\
&= |\langle f_1(\theta_1 - \bar{f}(\theta_1), O_i), \theta_1 - \theta^* \rangle - \langle f_1(\theta_2, O_i) - \bar{f}_1(\theta_2), \theta_2 - \theta^* \rangle| \\
&\leq \|\theta_1 - \theta^*\|_2 \left\|f_1(\theta_1, O_i) - \bar{f}_1(\theta_1) - f_1(\theta_2, O_i) + \bar{f}_1(\theta_2)\right\|_2 + \left\|f_1(\theta_2, O_i) - \bar{f}_1(\theta_2)\right\|_2 \|\theta_1 - \theta_2\|_2 \\
&\leq \|\theta_1 - \theta^*\|_2 (\|f_1(\theta_1, O_i) - f_1(\theta_2, O_i)\|_2 + \left\|\bar{f}_1(\theta_1) - \bar{f}_1(\theta_2)\right\|_2) + \left\|f_1(\theta_2, O_i) - \bar{f}_1(\theta_2)\right\|_2 \|\theta_1 - \theta_2\|_2 \\
&\leq 2R_\theta(\left\|(A_t - B_t C^{-1} A)(\theta_1 - \theta_2)\right\|_2 + \left\|(A - BC^{-1} A)(\theta_1 - \theta_2)\right\|_2) + 2K_{f_1}\|\theta_1 - \theta_2\|_2 \\
&\leq 4R_\theta(1 + \gamma)\rho_{\max}(1 + \frac{1}{\lambda_{cm}}\gamma\rho_{\max})\|\theta_1 - \theta_2\|_2 + 2K_{f_1}\|z_1 - z_2\|_2 \\
&\leq L_{f_1, \theta}\|\theta_1 - \theta_2\|_2.
\end{aligned}
$$

$\square$

**Lemma 15.** *For $i \leq \tau_\alpha$, $\mathbb{E}[\zeta_{f_1}(\theta_i, O_i)] \leq c_\alpha L_{f_1, \theta}(K_{f_1} + K_{g_1})\tau_\alpha$; and for $i > \tau_\alpha$, $\mathbb{E}[\zeta_{f_1}(\theta_i, O_i)] \leq 8R_\theta K_{f_1}\alpha_t + L_{f_1, \theta}(K_{f_1} + K_{g_1})\tau_\alpha \alpha_{i-\tau_\alpha}$.*

*Proof.* Applying the Lipschitz continuous property of $\zeta_{g_2}(z, O_i)$ and the inequality (29) in Lemma 4, it follows that

$$
|\zeta_{f_1}(\theta_i, O_i) - \zeta_{f_1}(\theta_{i-\tau}, O_i)| \leq L_{f_1, \theta}(K_{f_1} + K_{g_1})\|\theta_i - \theta_{i-\tau}\|_2 \leq L_{f_1, \theta}(K_{f_1} + K_{g_1}) \sum_{k=i-\tau}^{i-1} \alpha_k.
$$

Then, we need to provide an upper bound for $\mathbb{E}[\zeta_{f_1}(\theta_{i-\tau}, O_i)]$. We further define an independent $\theta'_{i-\tau}$ and $O'_i = (s'_i, a'_i, r'_i, s'_{i+1})$, which have the same marginal distributions as $\theta_{i-\tau}$ and $O_i$. Using Lemma 14 and following the steps similar to those in Lemma 4, we have

$$
\mathbb{E}[\zeta_{f_1}(\theta_{i-\tau}, O_i)] \leq |\mathbb{E}[\zeta_{f_1}(\theta_{i-\tau}, O_i)] - \mathbb{E}[\zeta_{f_1}(\theta'_{i-\tau}, O'_i)]| \leq 8R_\theta K_{f_1} m\rho^\tau.
$$

If $i \leq \tau_\alpha$, it follows that

$$\mathbb{E}[\zeta_{f_1}(\theta_i, O_i)] \leq \mathbb{E}[\zeta_{f_1}(\theta_0, O_i)] + L_{f_1,\theta}(K_{f_1} + K_{g_1}) \sum_{k=0}^{i-1} \alpha_k \leq L_{f_1,\theta}(K_{f_1} + K_{g_1}) i \alpha_0$$

$$\leq c_\alpha L_{f_1,\theta}(K_{f_1} + K_{g_1}) \tau_\alpha.$$

If $i > \tau_\alpha$, it follows that

$$\mathbb{E}[\zeta_{f_1}(\theta_i, O_i)] \leq \mathbb{E}[\zeta_{f_1}(\theta_{i-\tau_\alpha}, O_i)] + L_{f_1,\theta}(K_{f_1} + K_{g_1}) \sum_{k=i-\tau_\alpha}^{i-1} \alpha_k$$

$$\leq 8 R_\theta K_{f_1} m \rho^{\tau_\alpha} + L_{f_1,\theta}(K_{f_1} + K_{g_1}) \tau_\alpha \alpha_{i-\tau_\alpha}$$

$$\leq 8 R_\theta K_{f_1} \alpha_t + L_{f_1,\theta}(K_{f_1} + K_{g_1}) \tau_\alpha \alpha_{i-\tau_\alpha}.$$

$\square$

**Lemma 16.** *Fix $0 < \sigma < 1$, and let $\sigma_t = c_\alpha/(1+t)^\sigma$. Then*

$$\sum_{i=0}^{t} e^{\lambda_\theta \sum_{k=i+1}^{t} \alpha_k} \alpha_i \mathbb{E}[\zeta_{f_1}(\theta_i, O_i)]$$

$$\leq c_\alpha L_{f_1,\theta}(K_{f_1} + K_{g_1}) \tau_\alpha \frac{e^{-\lambda_\theta c_\alpha}}{-\lambda_\theta} e^{\frac{\lambda_\theta c_\alpha}{1-\sigma}[(1+t)^\sigma - (1+\tau_\sigma)^\sigma]} + 8 R_\theta K_{f_1} \frac{e^{-\lambda_\theta c_\alpha}}{-\lambda_\theta} \frac{c_\alpha}{(1+t)^\sigma}$$

$$+ L_{f_1,\theta}(K_{f_1} + K_{g_1}) \tau_\alpha \frac{2 e^{-\lambda_\theta c_\alpha/2}}{-\lambda_\theta} (e^{\frac{\lambda_\theta c_\alpha}{2(1-\sigma)}[(t+1)^{1-\sigma} - 1]} D_4 \mathbb{1}_{\{\tau_\alpha + 1 < i_\alpha\}} + \alpha_{t-\tau_\alpha}),$$

*where $T_n = \sum_{k=0}^{n-1} \alpha_k$, $D_4 = c_\alpha \max_{i \in [0, i_{d_4}]} \{ e^{-(\lambda_\theta/2) \sum_{k=0}^{i} \alpha_k} \}$ and $i_{d_4} = (\frac{-2\sigma}{\lambda_\theta c_\alpha})^{\frac{1}{1-\sigma}}$.*

*Proof.* Applying Lemma 15, it follows that

$$\sum_{i=0}^{t} e^{\lambda_\theta \sum_{k=i+1}^{t} \alpha_k} \alpha_i \mathbb{E}[\zeta_{f_1}(\theta_i, O_i)]$$

$$\leq c_\alpha L_{f_1,\theta}(K_{f_1} + K_{g_1}) \tau_\alpha \sum_{i=0}^{\tau_\alpha} e^{\lambda_\theta \sum_{k=i+1}^{t} \alpha_k} \alpha_i + 8 R_\theta K_{f_1} \alpha_t \sum_{i=\tau_\alpha+1}^{t} e^{\lambda_\theta \sum_{k=i+1}^{t} \alpha_k} \alpha_i$$

$$+ L_{f_1,\theta}(K_{f_1} + K_{g_1}) \tau_\alpha \sum_{i=\tau_\alpha+1}^{t} e^{\lambda_\theta \sum_{k=i+1}^{t} \alpha_k} \alpha_{i-\tau_\beta} \alpha_i. \tag{48}$$

Following steps similar to those in Lemma 5, we obtain:

$$\sum_{i=0}^{\tau_\alpha} e^{\lambda_w \sum_{k=i+1}^{t} \alpha_k} \alpha_i \leq \frac{e^{-\lambda_\theta c_\alpha}}{-\lambda_\theta} e^{\frac{\lambda_\theta c_\alpha}{1-\sigma}[(1+t)^\sigma - (1+\tau_\sigma)^\sigma]} \tag{49}$$

$$\alpha_t \sum_{i=\tau_\alpha+1}^{t} e^{\lambda_\theta \sum_{k=i+1}^{t} \alpha_k} \alpha_i \leq \frac{e^{-\lambda_\theta c_\alpha}}{-\lambda_\theta} \frac{c_\alpha}{(1+t)^\sigma} \tag{50}$$

$$\sum_{i=\tau_\alpha+1}^{t} e^{\lambda_\theta \sum_{k=i+1}^{t} \alpha_k} \alpha_{i-\tau_\alpha} \alpha_i \leq \frac{2 e^{-\lambda_\theta c_\alpha/2}}{-\lambda_\theta} (e^{\frac{\lambda_\theta c_\alpha}{2(1-\sigma)}[(t+1)^{1-\sigma} - 1]} D_4 \mathbb{1}_{\{\tau_\alpha + 1 < i_\alpha\}} + \alpha_{t-\tau_\alpha}), \tag{51}$$

which yields the desired result. $\square$

**Lemma 17.** *For $0 < \sigma < 1$, $c_\alpha > 0$, $\alpha_t = \frac{c_\alpha}{(1+t)^\sigma}$, and $0 < x < 1$, $0 < y < 1$. If $\mathbb{E} \|z_t\|_2^2 = \mathcal{O}(\frac{\log t}{t^\nu} + \frac{1}{t^\nu})^x$ and $\mathbb{E} \|\theta_t - \theta^*\|_2^2 = \mathcal{O}(\frac{\log t}{t^\nu} + \frac{1}{t^\nu})^y$ for $a, b > 0$, then we have*

$$\sum_{i=0}^{t} e^{\lambda_\theta \sum_{k=i+1}^{t} \alpha_k} \alpha_i \mathbb{E}[\langle B_i z_i, \theta_i - \theta^* \rangle] = \mathcal{O}\Big( \frac{\log t}{t^\nu} + \frac{1}{t^\nu} \Big)^{0.5(x+y)}.$$

*If* $\mathbb{E}\|z_t\|_2^2 = \mathcal{O}(\frac{\log t}{t^\nu} + \frac{1}{t^{2(\sigma-\nu)-\epsilon}})^x$, $\mathbb{E}\|\theta_t - \theta^*\|_2^2 = \mathcal{O}(\frac{\log t}{t^\nu} + \frac{1}{t^{2(\sigma-\nu)-\epsilon}})^y$, *then we have*

$$\sum_{i=0}^{t} e^{\lambda_\theta \sum_{k=i+1}^{t} \alpha_k} \alpha_i \mathbb{E}[\langle B_i z_i, \theta_i - \theta^* \rangle] = \mathcal{O}\Big( \frac{\log t}{t^\nu} + \frac{1}{t^{2(\sigma-\nu)-\epsilon}} \Big)^{0.5(x+y)}.$$

*Proof.* Consider the first case. Without loss of generality, we assume that there exist constant $0 < C_1, C_2 < \infty$, $T > 0$ such that

$$\mathbb{E}\|z_t\|_2^2 \le 4R_w^2 \qquad \text{for all } 0 \le t \le T,$$
$$\mathbb{E}\|z_t\|_2^2 \le C_1^2 \Big(\frac{\log t + 1}{t^\nu}\Big)^x \quad \text{for all } t > T,$$

and

$$\mathbb{E}\|\theta_t - \theta^*\|_2^2 \le R_\theta^2 \qquad \text{for all } 0 \le t \le T,$$
$$\mathbb{E}\|\theta_t - \theta^*\|_2^2 \le C_2^2 \Big(\frac{\log t + 1}{t^\nu}\Big)^y \quad \text{for all } t > T.$$

Then, it follows that

$$\sum_{i=0}^{t} e^{\lambda_\theta \sum_{k=i+1}^{t} \alpha_k} \alpha_i \mathbb{E}[\langle B_i z_i, \theta_i - \theta^* \rangle]$$

$$\le \sum_{i=0}^{t} e^{\lambda_\theta \sum_{k=i+1}^{t} \alpha_k} \alpha_i \sqrt{\mathbb{E}[\|B_i z_i\|_2^2]} \sqrt{\mathbb{E}[\|\theta_i - \theta^*\|_2^2]}$$

$$\le \sum_{i=0}^{t} e^{\lambda_\theta \sum_{k=i+1}^{t} \alpha_k} \alpha_i \|B_i\|_2 \sqrt{\mathbb{E}\|z_i\|_2^2} \sqrt{\mathbb{E}[\|\theta_i - \theta^*\|_2^2]}$$

$$\le \gamma\rho_{\max} \Big( \sum_{i=0}^{T} e^{\lambda_\theta \sum_{k=i+1}^{t} \alpha_k} \alpha_i \sqrt{\mathbb{E}\|z_i\|_2^2} \sqrt{\mathbb{E}[\|\theta_i - \theta^*\|_2^2]}$$

$$+ \sum_{i=T+1}^{t} e^{\lambda_\theta \sum_{k=i+1}^{t} \alpha_k} \alpha_i \sqrt{\mathbb{E}\|z_i\|_2^2} \sqrt{\mathbb{E}[\|\theta_i - \theta^*\|_2^2]} \Big)$$

$$\le \gamma\rho_{\max} \Big( 2R_w R_\theta \sum_{i=0}^{T} e^{\lambda_\theta \sum_{k=i+1}^{t} \alpha_k} \alpha_i + C_1 C_2 \sum_{i=T+1}^{t} e^{\lambda_\theta \sum_{k=i+1}^{t} \alpha_k} \alpha_i \Big( \frac{\log i + 1}{i^\nu} \Big)^{0.5(x+y)} \Big)$$

$$\le 2\gamma\rho_{\max} R_w R_\theta \Big( 2R_w R_\theta \frac{e^{-\lambda_\theta c_\alpha}}{-\lambda_\theta} e^{\frac{\lambda_\theta c_\alpha}{1-\sigma}[(1+t)^\sigma - (1+T)^\sigma]}$$

$$+ C_1 C_2 \frac{2e^{-\lambda_\theta C_\alpha/2}}{-\lambda_\theta} \Big( e^{\frac{\lambda_\theta c_\alpha}{2(1-\sigma)}[(1+t)^{1-\sigma} - 1]} D + \Big( \frac{\log t + 1}{t^\nu} \Big)^{0.5(x+y)} \Big) \Big)$$

Here we follow similar steps in (31) and (33)-(36) to obtain

$$\sum_{i=0}^{T} e^{\lambda_\theta \sum_{k=i+1}^{t} \alpha_k} \alpha_i \le \frac{e^{-\lambda_\theta c_\alpha}}{-\lambda_\theta} e^{\frac{\lambda_\theta c_\alpha}{1-\sigma}[(1+t)^\sigma - (1+T)^\sigma]},$$

and

$$\sum_{i=T+1}^{t} e^{\lambda_\theta \sum_{k=i+1}^{t} \alpha_k} \alpha_i \Big( \frac{\log i}{i^\nu} \Big)^{0.5(x+y)} \le \frac{2e^{-\lambda_\theta C_\alpha/2}}{-\lambda_\theta} \Big( e^{\frac{\lambda_\theta c_\alpha}{2(1-\sigma)}[(1+t)^{1-\sigma} - 1]} D + \Big( \frac{\log t + 1}{t^\nu} \Big)^{0.5(x+y)} \Big),$$

where $0 < D < \infty$ is a constant depend on $x$ and $y$.

The proof for the second case follows similarly. $\qquad\square$

**Lemma 18.** *For $0 < \frac{3}{2}\nu < \sigma < 1$, if $\mathbb{E}\|z_t\|_2^2 = \mathcal{O}(\frac{\log t}{t^\nu}) + \mathcal{O}(\frac{1}{t^\nu})$, then*

$$\sum_{i=0}^{t} e^{\lambda_\theta \sum_{k=i+1}^{t} \alpha_k} \alpha_i \mathbb{E}[\langle B_i z_i, \theta_i - \theta^* \rangle] = \mathcal{O}\Big( \frac{\log t}{t^\nu} + \frac{1}{t^\nu} \Big)^{1-\epsilon'},$$

*and for $0 < \nu < \sigma \leq \frac{3}{2}\nu < 1$, if $\mathbb{E}\|z_t\|_2^2 = \mathcal{O}(\frac{\log t}{t^\nu}) + \mathcal{O}(\frac{1}{t^{2(\sigma-\nu)-\epsilon}})$, then*

$$\sum_{i=0}^t e^{\lambda_\theta \sum_{k=i+1}^t \alpha_k} \alpha_i \mathbb{E}[\langle B_i z_i, \theta_i - \theta^*\rangle] = \mathcal{O}\Big(\frac{\log t}{t^\nu} + \frac{1}{t^{2(\sigma-\nu)-\epsilon}}\Big)^{1-\epsilon'},$$

*where $\epsilon'$ can be any constant in $(0, 0.5]$.*

*Proof.* Consider the first case. First, $\mathbb{E}\|\theta_t - \theta^*\|_2^2 \leq 4R_\theta^2 = \mathcal{O}(1)$, applying Lemma 17 we immediately have

$$\sum_{i=0}^t e^{\lambda_\theta \sum_{k=i+1}^t \alpha_k} \alpha_i \mathbb{E}[\langle B_i z_i, \theta_i - \theta^*\rangle] = \mathcal{O}\Big(\frac{\log t}{t^\nu} + \frac{1}{t^\nu}\Big)^{0.5}. \tag{52}$$

Then it follows that $\mathbb{E}\|\theta_{t+1} - \theta^*\|_2^2 = \mathcal{O}(\frac{\log t}{t^\nu})^{0.5}$. Then again applying Lemmas 17 and (52), we obtain

$$\sum_{i=0}^t e^{\lambda_\theta \sum_{k=i+1}^t \alpha_k} \alpha_i \mathbb{E}[\langle B_i z_i, \theta_i - \theta^*\rangle] = \mathcal{O}\Big(\frac{\log t}{t^\nu} + \frac{1}{t^\nu}\Big)^{0.75}. \tag{53}$$

Hence, following the steps in (53) for a total number $N = \lceil \log_2(\frac{1}{1-\epsilon'}) \rceil$ of times, we have

$$\sum_{i=0}^t e^{\lambda_\theta \sum_{k=i+1}^t \alpha_k} \alpha_i \mathbb{E}[\langle B_i z_i, \theta_i - \theta^*\rangle] = \mathcal{O}\Big(\frac{\log t}{t^\nu} + \frac{1}{t^\nu}\Big)^{1-\frac{1}{2^N}}$$

$$= \mathcal{O}\Big(\frac{\log t}{t^\nu} + \frac{1}{t^\nu}\Big)^{1-\epsilon'}.$$

The proof for the second case follows similarly. $\qquad\square$

**Lemma 19.** *Let $\alpha_t = \frac{1}{-\lambda_\theta(1+t)}$. Then*

$$\sum_{i=0}^t \mathbb{E}\zeta_{f_1}(\theta_i, O_i) \leq \frac{2L_{f_1,\theta}(K_{f_1} + K_{g_1})}{-\lambda_\theta}\tau_\alpha^2 + \frac{8R_\theta K_{f_1}}{-\lambda_\theta} + L_{f_1,\theta}(K_{f_1} + K_{g_1})\tau_\alpha \ln(1+t).$$

*Proof.* Applying Lemma (15), it follows that

$$\sum_{i=0}^t \mathbb{E}\zeta_{f_1}(\theta_i, O_i) = \sum_{i=0}^{\tau_\alpha} \mathbb{E}\zeta_{f_1}(\theta_i, O_i) + \sum_{i=\tau_\alpha+1}^t \mathbb{E}\zeta_{f_1}(\theta_i, O_i)$$

$$\leq \frac{L_{f_1,\theta}(K_{f_1} + K_{g_1})}{-\lambda_\theta}\tau_\alpha(1+\tau_\alpha) + \frac{8R_\theta K_{f_1}(t-\tau_\alpha)}{-\lambda_\theta(1+t)}$$

$$+ L_{f_1,\theta}(K_{f_1} + K_{g_1})\tau_\alpha \sum_{i=\tau_\alpha+1}^t \alpha_{i-\tau_\alpha}$$

$$\leq \frac{2L_{f_1,\theta}(K_{f_1} + K_{g_1})}{-\lambda_\theta}\tau_\alpha^2 + \frac{8R_\theta K_{f_1}}{-\lambda_\theta} + L_{f_1,\theta}(K_{f_1} + K_{g_1})\tau_\alpha \sum_{i=1}^{t-\tau_\alpha} \frac{1}{1+i}$$

$$\leq \frac{2L_{f_1,\theta}(K_{f_1} + K_{g_1})}{-\lambda_\theta}\tau_\alpha^2 + \frac{8R_\theta K_{f_1}}{-\lambda_\theta} + L_{f_1,\theta}(K_{f_1} + K_{g_1})\tau_\alpha \ln(1+t)$$

$\qquad\square$

**Lemma 20.** *Suppose $0 < x < 1$, $0 < y \leq 1$. If $\mathbb{E}\|z_t\|_2^2 = \mathcal{O}(\frac{\log t}{t^\nu} + \frac{1}{t^\nu})^x$, $\mathbb{E}\|\theta_t - \theta^*\|_2^2 = \mathcal{O}(\frac{\log t}{t^\nu} + \frac{1}{t^\nu})^y$, then we have*

$$\frac{1}{1+t}\sum_{i=0}^t \mathbb{E}[\langle B_i z_i, \theta_i - \theta^*\rangle] = \mathcal{O}\Big(\frac{\log t}{t^\nu} + \frac{1}{t^\nu}\Big)^{0.5(x+y)}.$$

*If $\mathbb{E}\|z_t\|_2^2 = \mathcal{O}(\frac{\log t}{t^\nu} + \frac{1}{t^{2(\sigma-\nu)-\epsilon}})^x$ and $\mathbb{E}\|\theta_t - \theta^*\|_2^2 = \mathcal{O}(\frac{\log t}{t^\nu} + \frac{1}{t^{2(\sigma-\nu)-\epsilon}})^y$, then we have*

$$\frac{1}{1+t}\sum_{i=0}^t \mathbb{E}[\langle B_i z_i, \theta_i - \theta^*\rangle] = \mathcal{O}\Big(\frac{\log t}{t^\nu} + \frac{1}{t^{2(\sigma-\nu)-\epsilon}}\Big)^{0.5(x+y)}.$$

*Proof.* Consider the first case. Similarly to the proof in (17), without loss of generality, we can assume there exist constants $0 < C_1, C_2 < \infty$ and $T > 0$ such that

$$\mathbb{E}\|z_t\|_2^2 \leq 4R_w^2 \qquad \text{for all } 0 \leq t \leq T,$$

$$\mathbb{E}\|z_t\|_2^2 \leq C_1^2\Big(\frac{\log t}{t^\nu} + \frac{1}{t^\nu}\Big)^x \qquad \text{for all } t > T,$$

and

$$\mathbb{E}\|\theta_t - \theta^*\|_2^2 \leq R_\theta^2 \qquad \text{for all } 0 \leq t \leq T,$$

$$\mathbb{E}\|\theta_t - \theta^*\|_2^2 \leq C2^2\Big(\frac{\log t}{t^\nu} + \frac{1}{t^\nu}\Big)^y \qquad \text{for all } t > T.$$

Then, it follows that

$$\frac{1}{1+t}\sum_{i=0}^{t}\mathbb{E}[\langle B_i z_i, \theta_i - \theta^*\rangle]$$

$$\leq \frac{1}{1+t}\sum_{i=0}^{t}\sqrt{\mathbb{E}[\|B_i z_i\|_2^2]}\sqrt{\mathbb{E}[\|\theta_i - \theta^*\|_2^2]}$$

$$\leq \frac{1}{1+t}\sum_{i=0}^{t}\|B_i\|_2\sqrt{\mathbb{E}\|z_i\|_2^2}\sqrt{\mathbb{E}[\|\theta_i - \theta^*\|_2^2]}$$

$$\leq \gamma\rho_{\max}\frac{1}{1+t}\Big(\sum_{i=0}^{T}\sqrt{\mathbb{E}\|z_i\|_2^2}\sqrt{\mathbb{E}[\|\theta_i - \theta^*\|_2^2]} + \sum_{i=T+1}^{t}\sqrt{\mathbb{E}\|z_i\|_2^2}\sqrt{\mathbb{E}[\|\theta_i - \theta^*\|_2^2]}\Big)$$

$$\leq \gamma\rho_{\max}\frac{1}{1+t}\Big(2R_w R_\theta(1+T) + C_1 C_2\sum_{i=T+1}^{t}\Big(\frac{\log i + 1}{i^\nu}\Big)^{0.5(x+y)}\Big)$$

$$\leq \gamma\rho_{\max}\frac{1}{1+t}\Big(2R_w R_\theta(1+T) + C_1 C_2(\log t + 1)^{0.5(x+y)}\sum_{i=T+1}^{t}\Big(\frac{1}{i^\nu}\Big)^{0.5(x+y)}\Big)$$

$$\leq \gamma\rho_{\max}\Big(2R_w R_\theta\frac{1+T}{1+t} + D\Big(\frac{\log t + 1}{t^\nu}\Big)^{0.5(x+y)}\Big),$$

where $0 < D < \infty$ is a constant dependent on $x$ and $y$. The proof for the second case follows similarly. □

**Lemma 21.** *Suppose $0 < \nu < \frac{2}{3}$, if $\mathbb{E}\|z_t\|_2^2 = \mathcal{O}(\frac{\log t}{t^\nu} + \frac{1}{t^\nu})$, then*

$$\frac{1}{1+t}\sum_{i=0}^{t}\mathbb{E}[\langle B_i z_i, \theta_i - \theta^*\rangle] = \mathcal{O}\Big(\frac{\log t}{t^\nu} + \frac{1}{t^\nu}\Big)^{1-\epsilon'},$$

*and suppose $\frac{2}{3} \leq \nu < 1$, if $\mathbb{E}\|z_t\|_2^2 = \mathcal{O}(\frac{\log t}{t^\nu}) + \mathcal{O}(\frac{1}{t^{2(1-\nu)-\epsilon}})$, then*

$$\frac{1}{1+t}\sum_{i=0}^{t}\mathbb{E}[\langle B_i z_i, \theta_i - \theta^*\rangle] = \mathcal{O}\Big(\frac{\log t}{t^\nu} + \frac{1}{t^{2(1-\nu)-\epsilon}}\Big)^{1-\epsilon'},$$

*where $\epsilon'$ can be any constant in $(0, 0.5]$.*

*Proof.* We proof this lemma by following similar steps in the proof of Lemma 18. □

# B    Proof of Theorem 2

From (12) and use the fact that $\beta_t = \beta$ for all $t > 0$, we have

$$\mathbb{E}\left\|z_{t+1}\right\|_2^2 \le (1 - |\lambda_w|\beta)^{1+t}\left\|z_0\right\|_2^2$$
$$+ 2\beta \sum_{i=0}^{t}(1 - |\lambda_w|\beta)^{t-i}[\zeta_{f_2}(\theta_i, z_i, O_i)]$$
$$+ 2\beta \sum_{i=0}^{t}(1 - |\lambda_w|\beta)^{t-i}[\zeta_{g_2}(z_i, O_i)]$$
$$+ 2 \sum_{i=0}^{t}(1 - |\lambda_w|\beta)^{t-i}\mathbb{E}\langle C^{-1}A(\theta_{i+1} - \theta_i), z_i\rangle$$
$$+ 3(K_{f_2}^2 + K_{g_2}^2)\beta^2 \sum_{i=0}^{t}(1 - |\lambda_w|\beta)^{t-i} + 3\eta^2\beta^2 K_{r_1}^2 \sum_{i=0}^{t}(1 - |\lambda_w|\beta)^{t-i}. \tag{54}$$

By slightly modifying the proof of Lemma 4, Lemma 8 and Lemma 10, we have

$$\mathbb{E}[\zeta_{f_2}(\theta_i, z_i, O_i)] \le \beta(8R_w K_{f_2} + K_{r_3}\tau_\beta), \tag{55}$$

and

$$\mathbb{E}[\zeta_{g_2}(z_i, O_i)] \le \beta(8R_w K_{g_2} + L_{g_2,z}K_{r_2}\tau_\beta), \tag{56}$$

and

$$\sum_{i=0}^{t}(1 - |\lambda_w|\beta)^{t-i}\mathbb{E}\langle C^{-1}A(\theta_{i+1} - \theta_i), z_i\rangle \le \frac{2(1 + \gamma)\rho_{\max}R_w(K_{g_1} + K_{f_1})}{c_\beta|\lambda_w|\lambda_{cm}}\eta \tag{57}$$

Substituting (55), (56) and (57) into (54), and use the fact that $\tau_\beta < \log_{\frac{1}{\rho}}\frac{m}{\rho} + \ln^{-1}(\frac{1}{\rho})\ln(\frac{1}{\beta})$, we have

$$\mathbb{E}\left\|z_{t+1}\right\|_2^2 \le (1 - |\lambda_w|\beta)^{1+t}\left\|z_0\right\|_2^2$$
$$+ \frac{2(K_{r_3} + L_{g_2,z}K_{r_2})}{|\lambda_w|}\left(\log_{\frac{1}{\rho}}\frac{m}{\rho} + \ln^{-1}(\frac{1}{\rho})\ln(\frac{1}{\beta})\right)\beta$$
$$+ \frac{[16R_w(K_{f_2} + K_{g_2}) + 3(K_{f_2}^2 + K_{g_2}^2) + 3\eta^2 K_{r_1}^2]}{|\lambda_w|}\beta$$
$$+ \frac{2(1 + \gamma)\rho_{\max}R_w(K_{g_1} + K_{f_1})}{|\lambda_w|\lambda_{cm}}\eta. \tag{58}$$

Let

$$C_5 = \frac{2(K_{r_3} + L_{g_2,z}K_{r_2})}{|\lambda_w|}\left(\log_{\frac{1}{\rho}}\frac{m}{\rho} + \ln^{-1}(\frac{1}{\rho})\right)$$
$$+ \frac{[16R_w(K_{f_2} + K_{g_2}) + 3(K_{f_2}^2 + K_{g_2}^2) + 3K_{r_1}^2]}{|\lambda_w|} \tag{59}$$

and

$$C_6 = \frac{2(1 + \gamma)\rho_{\max}R_w(K_{g_1} + K_{f_1})}{|\lambda_w|\lambda_{cm}} \tag{60}$$

then we have

$$\mathbb{E}\left\|z_t\right\|_2^2 \le (1 - |\lambda_w|\beta)^t\left\|z_0\right\|_2^2 + C_5 \max\{\beta, \ln(\frac{1}{\beta})\beta\} + C_6\eta.$$

Let $T = \lceil\frac{\ln[C_5\max\{\beta,\ln(\frac{1}{\beta})\beta\}/\|z_0\|_2^2]}{-\ln(1 - |\lambda_w|\beta)}\rceil$. Then

$$\mathbb{E}\left\|z_t\right\|_2^2 \le 4R_w^2 \qquad\qquad\qquad \text{for all } 0 \le t \le T, \tag{61}$$

$$\mathbb{E}\left\|z_t\right\|_2^2 \le 2C_5 \max\{\beta, \ln(\frac{1}{\beta})\beta\} + C_6\eta, \qquad \text{for all } t > 0. \tag{62}$$

Consider the recursion of $\theta_t$. From (20) and use the fact that $\alpha_t = c_\alpha \alpha$ for all $t > 0$, we have

$$\mathbb{E}\left\|\theta_{t+1} - \theta^*\right\|_2^2 \le (1 - |\lambda_\theta|\alpha)^{1+t}\left\|\theta_0 - \theta^*\right\|_2^2$$

$$+ 2\alpha\sum_{i=0}^{t}(1 - |\lambda_\theta|\alpha)^{t-i}\mathbb{E}[\zeta_{f_1}(\theta_i, O_i)] \tag{63}$$

$$+ 2\alpha\sum_{i=0}^{t}(1 - |\lambda_\theta|\alpha)^{t-i}\mathbb{E}\langle B_i z_i, \theta_i - \theta^*\rangle \tag{64}$$

$$+ 2(K_{f_1}^2 + K_{g_1}^2)\alpha^2\sum_{i=0}^{t}(1 - |\lambda_\theta|\alpha)^{t-i}.$$

By slightly modifying the proof of Lemma 15, we have

$$\mathbb{E}[\zeta_{f_1}(\theta_i, O_i)] \le \alpha(8R_\theta K_{f_1} + L_{f_1,\theta}(K_{f_1} + K_{g_1})\tau_\alpha). \tag{65}$$

Substitute (61) and (62) into (64), we have

$$2\alpha\sum_{i=0}^{t}(1 - |\lambda_\theta|\alpha)^{t-i}\mathbb{E}\langle B_i z_i, \theta_i - \theta^*\rangle$$

$$\le 4\alpha\gamma\rho_{\max}R_\theta\left[2R_w\sum_{i=0}^{T}(1 - |\lambda_\theta|\alpha)^{t-i} + (2C_5\max\{\beta, \ln(\frac{1}{\beta})\beta\} + C_6\eta)^{0.5}\sum_{i=T+1}^{t}(1 - |\lambda_\theta|\alpha)^{t-i}\right]$$

$$\le 8\gamma\rho_{\max}R_\theta R_w\frac{1 - (1 - |\lambda_\theta|\alpha)^{T+1}}{|\lambda_\theta|(1 - |\lambda_\theta|\alpha)^T}(1 - |\lambda_\theta|\alpha)^t + \frac{4\gamma\rho_{\max}R_\theta}{|\lambda_\theta|}(2C_5\max\{\beta, \ln(\frac{1}{\beta})\beta\} + C_6\eta)^{0.5} \tag{66}$$

Substitute (65) and (66) into (63) and (64) and using the fact that $\tau_\alpha < \log_{\frac{1}{\rho}}\frac{m}{\rho} + \ln^{-1}(\frac{1}{\rho})\ln(\frac{1}{\alpha})$ we have

$$\mathbb{E}\left\|\theta_{t+1} - \theta^*\right\|_2^2 \le (1 - |\lambda_\theta|\alpha)^{1+t}\left\|\theta_0 - \theta^*\right\|_2^2$$

$$+ \frac{2L_{f_1,\theta}(K_{f_1} + K_{g_1})}{|\lambda_\theta|}(\log_{\frac{1}{\rho}}\frac{m}{\rho} + \ln^{-1}(\frac{1}{\rho})\ln(\frac{1}{\alpha}))\alpha$$

$$+ \frac{2c_\alpha(8R_\theta K_{f_1} + K_{f_1}^2 + K_{g_1}^2)}{|\lambda_\theta|}\alpha$$

$$+ \frac{4\gamma\rho_{\max}R_\theta}{|\lambda_\theta|}(2C_5\max\{\beta, \ln(\frac{1}{\beta})\beta\} + C_6\eta)^{0.5}$$

$$+ 8\gamma\rho_{\max}R_\theta R_w\frac{1 - (1 - |\lambda_\theta|\alpha)^{T+1}}{|\lambda_\theta|(1 - |\lambda_\theta|\alpha)^T}(1 - |\lambda_\theta|\alpha)^t.$$

Let

$$C_2 = \frac{2L_{f_1,\theta}(K_{f_1} + K_{g_1})}{|\lambda_\theta|}(\log_{\frac{1}{\rho}}\frac{m}{\rho} + \ln^{-1}(\frac{1}{\rho})) + \frac{2(8R_\theta K_{f_1} + K_{f_1}^2 + K_{g_1}^2)}{|\lambda_\theta|}, \tag{67}$$

and

$$C_3 = 32\left(\frac{\gamma\rho_{\max}R_\theta}{|\lambda_\theta|}\right)^2 C_5, \tag{68}$$

and

$$C_4 = 16\left(\frac{\gamma\rho_{\max}R_\theta}{|\lambda_\theta|}\right)^2 C_6, \tag{69}$$

then we have

$$\mathbb{E}\left\|\theta_{t+1} - \theta^*\right\|_2^2 \le (1 - |\lambda_\theta|\alpha)^{1+t}(\left\|\theta_0 - \theta^*\right\|_2^2 + C_1) + C_2\max\{\alpha, \ln(\frac{1}{\alpha})\alpha\}$$

$$+ (C_3\max\{\beta, \ln(\frac{1}{\beta})\beta\} + C_4\eta)^{0.5} \tag{70}$$

where $C_1 = 8\gamma\rho_{\max}R_\theta R_w\frac{1 - (1 - |\lambda_\theta|\alpha)^{T+1}}{|\lambda_\theta|(1 - |\lambda_\theta|\alpha)^{T+1}}$.

## C  Proof of Theorem 3

We define vector $x_t = [\theta_t^\top, w_t^\top]^\top$ and $x^* = [\theta^{*\top}, 0^\top]^\top$, convex set $X = \{x | \sum_{i=1}^{d} x_i^2 \leq R_\theta^2$ and $\sum_{i=d+1}^{2d} x_i^2 \leq R_w^2\}$ and the projection operator $\Pi_X(x) = \operatorname{argmin}_{x':x' \in X} ||x - x'||_2$. We also define

$$G_t = \begin{bmatrix} A_t & B_t \\ \eta A_t & \eta B_t \end{bmatrix}, \quad g_t = \begin{bmatrix} b_t \\ \eta b_t \end{bmatrix},$$

and

$$G = \begin{bmatrix} A & B \\ \eta A & \eta B \end{bmatrix}, \quad g = \begin{bmatrix} b \\ \eta b \end{bmatrix}.$$

Then, we can rewrite the update of (1)-(2)

$$x_{t+1} = \Pi_X(x_t + \alpha_t(G_t x_t + g_t)). \tag{71}$$

We define $h(x_t, O_t) = G_t x_t + g_t$ and $\bar{h}(x_t) = G x_t + g$. Then, for the recursion of $x_t$ in (71), for any $t > 0$, we have

$$\begin{aligned}
||x_{t+1} - x^*||_2^2 &= ||\Pi_X(x_t + \alpha_t h(x_t, O_t)) - x^*||_2^2 \\
&= ||\Pi_X(x_t + \alpha_t h(x_t, O_t)) - \Pi_X(x^*)||_2^2 \\
&\leq ||x_t - x^* + \alpha_t h(x_t, O_t)||_2^2 \\
&= ||x_t - x^*||_2^2 + 2\alpha_t \langle h(x_t, O_t), x_t - x^* \rangle + \alpha_t^2 ||h(x_t, O_t)||_2^2 \\
&= ||x_t - x^*||_2^2 + 2\alpha_t \langle \bar{h}(x_t), x_t - x^* \rangle + 2\alpha_t \langle h(x_t, O_t) - \bar{h}(x_t), x_t - x^* \rangle + \alpha_t^2 ||h(x_t, O_t)||_2^2 \\
&= (1 - \alpha_t |\lambda_x|) ||x_t - x^*||_2^2 + 2\alpha_t \zeta_h(x_t, O_t) + \alpha_t^2 ||h(x_t, O_t)||_2^2, \tag{72}
\end{aligned}$$

where $\lambda_x = \lambda_{\max}(G + G^\top)$, and $\lambda_x < 0$ as shown in [17]. Then, consider the update in any block $s > 0$. Taking expectation on both sides conditional on the filtration $\mathcal{F}_{s-1}$ up to block $s-1$ and telescoping (71) yield that

$$\begin{aligned}
\mathbb{E}[||x_s - x^*||_2^2 | \mathcal{F}_{s-1}] &\leq (1 - |\alpha_s||\lambda_x|)^{T_s} ||x_{s-1} - x^*||_2^2 \\
&\quad + 2\alpha_s \sum_{i=1}^{T_s} (1 - \alpha_s |\lambda_x|)^{T_s - i} \mathbb{E}[\zeta_h(x_{t_{s-1}+i}, O_{t_{s-1}+i})] \\
&\quad + \alpha_s^2 \sum_{i=1}^{T_s} (1 - \alpha_s |\lambda_x|)^{T_s - i} ||h(x_{t_{s-1}+i}, O_{t_{s-1}+i})||_2^2. \tag{73}
\end{aligned}$$

Following similar steps in the proof for Theorem 1, we have the following results:

(a) There exist constant $C_G$ and $C_g$ such that $||G_t||_2, ||G||_2 \leq C_G$ and $||g_t||_2, ||g||_2 \leq C_g$.

(b) For all $i > 0$, $||h(x_i, O_i)||_2 \leq K_h$, where $K_h = C_G \sqrt{R_\theta^2 + R_w^2} + C_g$.

(c) For all $i > 0$, $||\zeta_h(x_i, O_i)||_2 \leq 4 K_h \sqrt{R_\theta^2 + R_w^2}$.

(d) For all $i > 0$ and $x, x' \in X$, $||\zeta_h(x, O_i) - \zeta_h(x', O_i)||_2 \leq L_h ||x - x'||_2$, where $L_h = 4 C_G \sqrt{R_\theta^2 + R_w^2} + 2 K_h$.

(e) For all $i > 0$, $\mathbb{E}[\zeta_h(x_i, O_i)] \leq \alpha_s (8 K_h \sqrt{R_\theta^2 + R_w^2} + L_h K_h \tau_{\alpha_s})$.

Then, substituting (e) into (73), we obtain

$$\mathbb{E}[||x_s - x^*||_2^2 | \mathcal{F}_{s-1}] \leq (1 + \alpha_s \lambda_x)^{T_s} ||x_{s-1} - x^*||_2^2 + \frac{2}{|\lambda_x|} \alpha_s (8 K_h \sqrt{R_\theta^2 + R_w^2} + L_h K_h \tau_{\alpha_s}) + \frac{1}{|\lambda_x|} \alpha_s K_h^2.$$

Recall that $\tau_{\alpha_s} \leq \log_{\frac{1}{\rho}} \frac{m}{\rho} + \ln^{-1}(\frac{1}{\rho}) \ln(\frac{1}{\alpha_s})$. Then, we have

$$\mathbb{E}[||x_s - x^*||_2^2 | \mathcal{F}_{s-1}] \leq (1 + \alpha_s \lambda_x)^{T_s} ||x_{s-1} - x^*||_2^2 + C_7 \max\{\alpha_s, \ln(\frac{1}{\alpha_s}) \alpha_s\}, \tag{74}$$

where

$$C_7 = \frac{2}{|\lambda_x|}(8K_h\sqrt{R_\theta^2 + R_w^2} + L_hK_h\log_{\frac{1}{\rho}}\frac{m}{\rho} + L_hK_h\ln^{-1}(\frac{1}{\rho}) + \frac{1}{2}K_h^2). \tag{75}$$

Since $\max\{\alpha_s, \ln(\frac{1}{\alpha_s})\alpha_s\} \le \epsilon_{s-1}/(4C_7)$ and $(1 + \alpha_s\lambda_x)^{T_s} \le 1/4$, we have

$$\mathbb{E}[\|x_s - x^*\|_2^2 \,|\mathcal{F}_{s-1}] \le \frac{1}{2}\epsilon_{s-1}.$$

After $S = \lceil\log_2(\epsilon_0/\epsilon)\rceil$ blocks we have

$$\mathbb{E}\,\|\theta_S - \theta^*\|_2^2 \le \mathbb{E}\,\|x_S - x^*\|_2^2 \le \epsilon.$$

The total iteration complexity is $\sum_{s=1}^{S} T_s = \mathcal{O}(\frac{1}{\epsilon}\log\frac{1}{\epsilon})$.