[Reviews · NeurIPS 2019]

Reviewer 1



This paper analyzes the non-asymptotic convergence for two time-scale TDC under a non-i.i.d. Markovian sample path and linear function approximation. The results are new and important to the field, and the analysis in this setting seems nontrivial. In addition, the paper also develops a new variant of TDC under a blockwise diminishing stepsize, and proves it asymptotically convergent with an arbitrarily small training error at linear convergence rate. Extensive experiments demonstrate that the new TDC variant can converge as fast as vanilla TDC with constant stepsize, and at the same time it enjoys comparable accuracy as TDC with diminishing stepsize. Overall, the paper has both analytical as well as practical value. However, the following issues need to be addressed. 1. The non-asymptotic convergence analysis for other GTD algorithms under a non-i.i.d. Markovian sample path has been studied in e.g., [30,34]. Hence, the new challenges of analyzing TDC relative to GTD in [30, 34] need to be compared and highlighted. 2. The paper generalizes the stagewise stepsize in the conventional (one timescale) optimization e.g., [34] to the considered two-timescale optimization. The new challenges of analyzing algorithms with blockwise diminishing stepsize in this new settings need to be discussed. 3. It is mentioned that the non-asymptotic analysis can be applied to studying other off-policy algorithms such as the actor-critic and the gradient Q-learning algorithms. A comment is due on how the theoretical guarantees can be affected in these settings?

Reviewer 2



This paper provides finite-time bounds for TD with gradient correction (TDC). While non-asymptotic behavior of TD and asymptotic behavior of TDC have been studied before, non-asymptotic analysis for TDC is new and interesting given the importance of off-policy learning and the challenge of step-size tuning in two time-scale algorithms. The paper is well-written, the discussion on the impact of the two step-sizes is clear and is supported by experiments. Questions: - The plots show the error between \theta and \theta^*. How is \theta^* obtained for these domains? - How would the worst-case errors predicted by the bound compare to the errors observed empirically in the experiments? - Besides implications for the choice of step-size, do these bounds provide insight on what properties of the problem, the behavior policy, and the representation affect the rate of convergence? - The first paragraph says that gradient-based TD algorithms are "more flexible than on-policy learning in practice." What does more flexible mean here?

Reviewer 3



To my knowledge the proposed analysis about two time-scale TDC under diminishing step-size and constant step-size is novel. However I need to acknowledge that I am not familiar with all related work in this area and I did not go through the proof details. Later, an blockwise diminishing step-size method is proposed to combine the advantage of constant step-size and diminishing step-size. However, to me, it looks like this ideal property need an careful choice of blocksize, and Thm 3 seems verified that. I have some concern about how to set the blocksize properly here without prior knowledge and how robust the algorithm is with respect to the blocksize hyperparameter. === After Rebuttal === I'm glad that the authors could show the algorithm performance is very robust to the blocksize. The author's further explanation addressed my concern. So I will change my score accordingly.

[Author Response · NeurIPS 2019]

**Reviewer 1: Q:** Compare and highlight new challenges of analyzing TDC relative to other GTD algorithms in [30,34].

**A:** (a) As mentioned in [14], TDC does not have an explicit saddle point representation as GTD and GTD2, and hence its analysis cannot follow the convex-concave optimization framework developed in [30,34]. (b) [30,34] assume that two variables' updates have the same constant stepsize. For TDC, we analyze more general cases: two stepsizes have different diminishing rates, and two stepsizes are different valued constants. Consequently, in our case, interaction between two variables requires more sophisticated techniques to analyze, e.g., recursively sharpening error bounds.

**Q:** The paper generalizes stagewise stepsize in conventional (one timescale) optimization to two-timescale optimization. Discuss new challenges of analyzing algorithms with blockwise diminishing stepsize in this new settings.

**A:** Comparing to conventional optimization with stagewise stepsize, here we need to handle the bias induced by non-i.i.d. samples and characterize non-asymptotic behavior of two timescale variable update. Hence, the update scheme for stepsize and block length in each block is designed based on two time-scale analysis to yield linear convergence rate blockwisely and desirable sample complexity.

**Q:** How the theoretical guarantees can be affected in the non-asymptotic analysis of actor-critic and gradient Q-learning.

**A:** Since both actor-critic and gradient Q-learning algorithms are two time-scale algorithms, our non-asymptotic analysis for two time-scale algorithms can be very useful. Moreover, analysis of these two algorithms will further require to deal with their special structures such as policy update, presence of multiple fixed points, local convergence, etc.

**Reviewer 2: Q:** How is $\theta^*$ obtained in the experiments.

**A:** In our experiment (Garnet problem), since we pick behavior policy $\pi_b$ and transition probability $p(s'|s, a)$, the stationary distribution $\mu_{\pi_b}$ can be computed. Since we also know target policy $\pi$ and feature matrix $\Phi$, we can compute the matrix $A$ and the vector $b$ by definition to obtain $\theta^* = -A^{-1}b$. Alternatively, $\theta^*$ can also be estimated by running the algorithm with diminishing stepsize for sufficiently long time and taking the average of outputs of several runs.

**Q:** How would worst-case errors predicted by the bound compare to errors observed empirically in experiments.

**A:** Our theory captures how the error bound changes with stepsize diminishing parameters $(\nu, \sigma)$, which agrees with how the empirical error changes with stepsize diminishing parameters in our experiments (see Fig. 1 in the paper). Furthermore, specializing Theorem 1 to i.i.d. scenarios, our convergence rate order-wisely matches the best known result in [Dalal et al. COLT 2018]. It is a good idea to plot theoretical errors and compare with empirical bounds. The main challenge here is that precisely estimating some parameters in the error bound (e.g., eq (25)) can be difficult (although they are known to be constants in convergence analysis). For example, mixing time parameters $\tau_\alpha$ and $\tau_\beta$ in (25) depend on geometric ergodicity of Markov chain, but constants $m$ and $\rho$ (see Assumption 3) are usually difficult to estimate in practice. We are currently further exploring such an issue.

**Q:** Besides implications for the choice of step-size, do these bounds provide insight on what properties of the problem, the behavior policy, and the representation affect the rate of convergence?

**A:** Theorem 1 (more precisely eq (25) in supple.) captures how other properties (besides stepsize) affect convergence rate. For example, convergence rate depends on $\lambda_\theta$, which is lower-bounded by the largest eigenvalue of matrix $2A^\top C^{-1}A$, and such a matrix is determined by behavior policy $\pi_b$, target policy $\pi$, transition probability $p(s'|s, a)$ and feature matrix $\Phi$. Convergence rate also depends on the mixing time $\tau_\alpha$ due to geometric ergodicity of the Markov chain, which is determined by $\pi_b$ and $p(s'|s, a)$. Other constant terms in (25) such as $L_{f_1,\theta}$, $K_{f_1}$ and $K_{g_1}$ capture the dependence on $\pi_b$, $\pi$, $\Phi$ and the discount factor $\gamma$.

**Q:** Explain what "more flexible" mean when saying gradient TD are "more flexible than on-policy learning in practice."

**A:** We meant gradient TD algorithms are flexible because they converge even with off-policy data and hence can exploit abundant samples (obtained under behavior policies) for learning when the on-policy samples are limited.

**Reviewer 3: Q:** How to set blocksize properly without prior knowledge and how robust the algorithm is with respect to blocksize hyperparameter.

A: In practice, blocksize $T_s$ and stepsize $\alpha_s$ are set by parameter tuning, but we do not directly tune them for all blocks because there are too many tuning parameters this way. Instead, Theorem 3 indicates that $T_s$ and $\alpha_s$ for all blocks are fully determined by only four parameters $\epsilon_0$, $|\lambda_x|$, $C_7$, and $\eta$, and among them $|\lambda_x|$ and $\eta$ can be estimated by matrices $A$ and $C$ from samples. Hence

we mainly tune only $\epsilon_0$ and $C_7$. Our experiments demonstrate that this approach yields desirable performance. For robustness, we run experiments (see the figure on the right) and find that perturbing blocksize even by $\pm30\%$ for all blocks changes the convergence rate only very slightly, demonstrating that the performance of algorithm is very robust to blocksize.

[Meta-Review · NeurIPS 2019]

All the reviewers recommended acceptance and, after consideration by the Senior AC and the Program Chairs, a recommendation for Accept (Poster) was settled on. [This meta-review was reviewed and revised by the Program Chairs]